# Towards A More Transparent Understanding of Weight-Averaged Model Merging: A Qualitative and Quantitative Study

## Abstract

Model merging, particularly through weight averaging, has shown surprising effectiveness in saving computations and improving model performance without any additional training. However, the interpretability of this technique works remains unclear. In this work, we reinterpret weight-averaged model merging through the lens of interpretability and provide empirical insights. We approach the problem from three perspectives: (1) we analyze the learned weight structures and demonstrate that model weights encode structured representations that help explain the compatibility of weight averaging; (2) we compare averaging in weight space and feature space across diverse model architectures (CNNs and ViTs) and datasets, aiming to expose under which circumstances what combination paradigm will work more effectively; (3) we study the effect of parameter scaling on prediction stability, highlighting how weight averaging acts as a form of regularization that contributes to robustness. By framing these analyses in an interpretability context, our work contributes to a more transparent and systematic understanding of model merging for stakeholders interested in the safety and reliability of untrained model combination methods. The code is available at `https://anonymous.4open.science/r/Rethink-Merge-E9BE`.

## 1 Introduction

Model merging combines multiple independently trained models into a single model, often without additional inference cost. This approach has been applied in a wide range of domains, including natural language processing Yang et al. (2024a), computer vision Li et al. (2023); Wang et al. (2025), and specific subfields such as federated learning Wang et al. (2020), knowledge distillation Khanuja et al. (2021), adversarial robustness Zhang et al. (2024); Croce et al. (2023), and large language model alignment Rame et al. (2024); Zhou et al. (2024a). Its potential to improve performance, increase efficiency, and offer greater deployment flexibility makes model merging an attractive paradigm. However, a critical question remains: from the standpoint of AI explainability and alignment, under what conditions the weight-averaged model merging can be beneficial for practice and is it reliable or predictable? Despite its growing practical use, the behavior of this technique has not been systematically characterized.

This research gap motivates the need to empirically characterize when and how the family of weight-averaged model-merging Wortsman et al. (2022) works. In this paper, we take a first step towards such an understanding by conducting a systematic qualitative and quantitative study from three perspectives:

- We relate the structured patterns encoded in model weights to the effectiveness of model merging. To the best of our knowledge, we are the first to build this connection for model merging through examining weight patterns in both linear classifiers and deep models across multiple datasets. The results show that the weights exhibit clear, structured organization, indicating that weight averaging functions as a meaningful linear combination of the individual weight vectors.

- We present a structured comparison between weight averaging and feature averaging, focusing on their respective roles in model merging and ensembling. Through analysis across diverse architectures

(including CNNs and ViTs) and datasets, we observe consistent behavioral differences between the two strategies, providing a more detailed understanding of how model merging and ensembling behave under different design choices.

- We systematically analyze how differences in weight magnitude and variance affect the robustness of model-merging predictions. Our results show that model merging implicitly serves as a form of regularization, favoring smoother and more stable parameter configurations. These results suggest that weight averaging can act as an implicit form of regularization, and provide evidence for when this behavior is beneficial in practice.

**The intention of the paper:** Instead of providing a thorough evaluation of existing works as a survey paper, this paper focuses on an empirical investigation of why and how weight-averaged model merging Wortsman et al. (2022) can be effective, through the three interpretability-driven perspectives outlined above. Importantly, we do not claim to give a complete mechanistic explanation of model merging. Rather, our goal is to provide interpretable empirical evidence that clarify and partially explain its behavior. Grounded in our empirical findings, we further provide practical proposals and insights that aim to guide future work and support a more transparent understanding of model merging within the broader AI alignment context.

## 2 Related Work

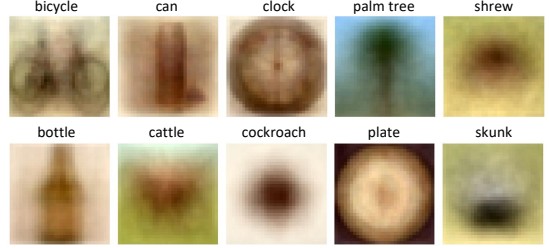

(a) Average of all images for each class.

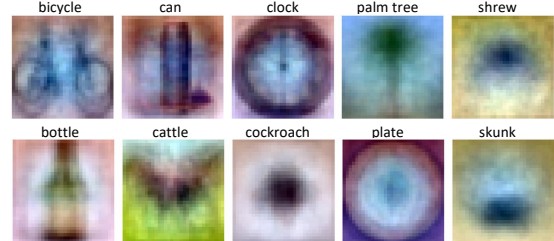

(b) Linear classifiers visualization.

Figure 1: Class-wise average images (a) and corresponding linear classifier visualizations (b) on the CIFAR-100 Krizhevsky (2009) dataset. A single linear classifier is trained across all classes, and each weight vector is reshaped to the input image dimensions for visualization. Detailed analysis is provided in the experimental section.

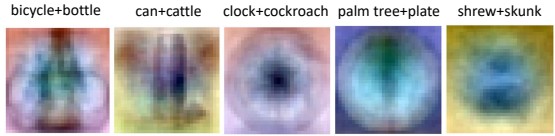

Figure 2: We merge the linear classifiers for the first 5 and last 5 classes (among the 10 selected) on CIFAR-100. This illustrates why model merging can be effective: it operates directly on the class-specific templates encoded in the model weights. Related analyses for deep models are provided in the supplementary materials.

In recent years, model-merging Sung et al. (2023); Sanjeev et al. (2024); Daheim et al. (2024); Navon et al. (2024); Singh & Jaggi (2020); Yang et al. (2024b); Jang et al. (2024); Almakky et al. (2024); Singh et al. (2024) has attracted growing interest. This approach has led to a range of methods focused on merging weights from independently trained models to enhance model performance or generalization. For example, Wortsman et al. (2022) proposed an approach of averaging, or greedily averaging, the parameters from various models to yield "Model Soups" that show more accurate performance than the original models used to produce the "soup" but without introducing more computations into inference. Further, Yadav et al. proposed Ties-Merging Yadav et al. (2024) based on parameter averaging by considering conflicts between the model weights, allowing for smooth parameter merging across models. Some other approaches Ainsworth

et al. (2023); Entezari et al. (2022); Jordan et al. (2023) address the problem from a different perspective, focusing on taking weight merging operations via finding a good way to interpolate on the loss basins. Zipit Stoica et al. (2024) takes one step ahead from Git Re-Basin Ainsworth et al. (2023) to align weights from inter-model and intra-model based on the alignment of features between merging models. Yu et al. Yu et al. (2024) proposed a DARE model to combine models trained on domain-specific data to create a domain-agnostic model-merging that generalizes well across diverse distributions. This approach ensures that domain knowledge is preserved while enhancing the model's ability to generalize across different tasks. Beyond model-merging, other techniques, such as mode connectivity path ensembles Garipov et al. (2018), analyze pathways in the parameter space between models. By identifying optimal modes along these paths, these models help improve the performance after merging. Fisher Merging Matena & Raffel (2022) selects parameters that approximately maximize the joint likelihood of the models' parameter posteriors.

Despite these advances, research into the empirical characters for practice that make model-merging effective remains limited. This gap in understanding serves as the motivation for our paper. Our study focuses on vanilla weight averaging without any alignment or reweighting, and thus can be viewed as characterizing the baseline regime that advanced merging methods try to improve upon.

In terms of the relation to mode connectivity and loss-landscape analyses Garipov et al. (2018); Entezari et al. (2022); Singh et al. (2024), they show that independently trained models can often be connected by low-loss curves in parameter space. Our empirical results can be seen as complementary: when models exhibit highly aligned templates across seeds (Section 3.3), uniform averaging corresponds to a point along such a low-loss path and tends to work well; when templates disagree, the average can land in a high-loss region, motivating more sophisticated merging schemes.

# 3    The Patterns Contained in Model Weights

In this section, we explain why weight-averaged model merging works by analyzing model weights through the lens of template matching. As illustrated in Fig. 1 and Fig. 2 for a linear model (with additional analyses for nonlinear models in the supplementary materials), we interpret weights as data-driven templates. This perspective connects naturally to the inner product operation, which underlies many neural computations. Viewing weights as templates helps make weight-space operations more interpretable, as they can be seen as manipulating feature representations directly.

## 3.1    View Model-merging from Template Matching Perspective

### 3.1.1    Linear Model Scenarios

The weights of a model can be interpreted as templates or prototypes associated with class-specific activations. In a linear classification, once the weight matrix is learned, we wrap it back to have the same size as the input images. Take CIFAR-100 for instance, each image within the dataset is $32{\times}32{\times}3$, if we flatten it and it will become a $3072{\times}1$ vector; the weight matrix size is $100{\times}3072$ without considering bias; in this case, each row of the weight matrix corresponds to each class to be classified and thus each row vector ($1{\times}3072$) can be wrapped back to $32{\times}32{\times}3$ with the same size of the original images. The learned weight matrix serves as simplified templates that represent each class within the data, as shown in Fig. 1. The extension of the weight template to non-linear scenarios can be found in the supplementary materials.

### 3.1.2    Non-linear Model Scenarios

To examine the extension of the template matching interpretation, we also visualize the convolutional kernels are shown in Fig. 3 for non-linear deep models ResNet50 on CIFAR-100. From the figure, we can observe that Kernel 5 is similar to Kernel 9, exhibiting comparable textures and texture orientations. However, Kernel 4 and Kernel 13 differ in both texture and texture orientation, which leads them to extract distinct features. For more non-linear model kernel examination, please refer to the appendix.

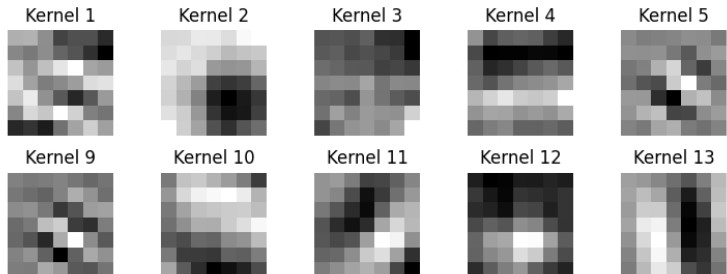

Figure 3: Kernel examination of ResNet50 on CIFAR-100 dataset.

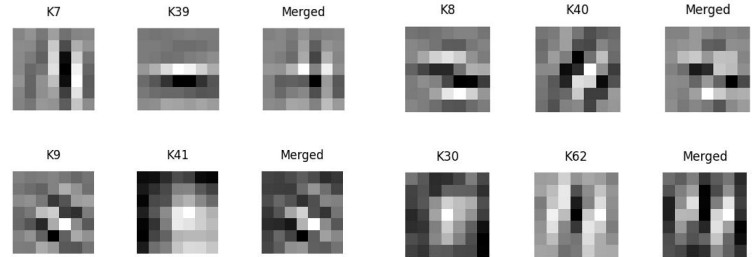

Figure 4: Merged Kernels of ResNet50 on the CIFAR-100 dataset.

We further examine the pair-wise merged kernels of ResNet50 on the CIFAR-100 dataset as shown in Fig. 4. We can clearly observe that the merged kernels exhibit the structural information from the kernels before merging.

### 3.2 Interpreting Model Merging Through the Lens of Template Matching

A key mechanism enabling these templates to operate effectively is the use of inner products to apply them to the input data. When an image is presented to the model, its activation is determined by the inner product between the image feature vector and per class weight vector. For each class, the weight vector that yields the highest activation is selected as the predicted class, indicating that the "template" of the current class is well-aligned with the input image. This phenomenon also holds for deeper networks. Weight templates in each layer are learned to detect different features that contribute to the objective functions (e.g., in shallow layers, low-level features tend to be learned; while in deeper layers, more semantically related features will be captured). The averaging operation in weight space, illustrated in Fig. 2, can be interpreted as a Mixup Zhang (2018)-like operation applied to model parameters. This suggests that linear combinations in weight space may produce effects analogous to those observed in feature space, blending learned representations in a structured and interpretable manner. This also relates to recent work Dekoninck et al. (2024); Zhang et al. (2023); Ilharco et al. (2023); Zhou et al. (2024b); Ortiz-Jimenez et al. (2023) showing that performing "model arithmetic" in weight space can yield effects analogous to feature arithmetic Mikolov (2013).

When extended to model merging, the combined model retains diverse patterns learned by each individual model. By representing a broader set of learned templates, the merged model is more likely to provide a good match for a given input, leading to improved performance over the individual components.

Since the inner product used in linear classification serves as a similarity measure, each row of the weight matrix $\mathbf{w} \in \mathbb{R}^n$ can be interpreted as a class-specific template. Given an input vector $\mathbf{x} \in \mathbb{R}^n$, the inner product $\mathbf{w}^\top \mathbf{x}$ quantifies the alignment between the input and the corresponding template: $\mathbf{w} \cdot \mathbf{x} = \sum_{i=1}^n w_i x_i = \|\mathbf{w}\|\|\mathbf{x}\| \cos\theta$, where $\|\mathbf{w}\|$ and $\|\mathbf{x}\|$ are the norms of $\mathbf{w}$ and $\mathbf{x}$, and $\theta$ is the angle between the two vectors. When $\theta = 0$, we have $\cos\theta = 1$, implying that $\mathbf{w} \cdot \mathbf{x} = |\mathbf{w}||\mathbf{x}|$. In this case, the weight vector $\mathbf{w}$ is perfectly aligned with the input $\mathbf{x}$, and the inner product is maximized. This corresponds to a perfect match between the template and the input. On the other hand, when $\theta \neq 0$, we have $\cos\theta < 1$,

indicating that the template and input features are misaligned. This principle extends to the layers within deep learning models, where alignment between learned weights and input representations similarly affects activation strength. Analyses for deep models are provided in the supplementary materials.

> **Remark 1**: The above analyses, together with the experimental results, indicate that weight matrices encode generalizable template information. When the performance of a merged model is suboptimal, it may be due to conflicts between the individually learned patterns, suggesting that resolving such inconsistencies in the weight space is critical for effective merging.

## 4  Averaging on Weights vs. Averaging on Features

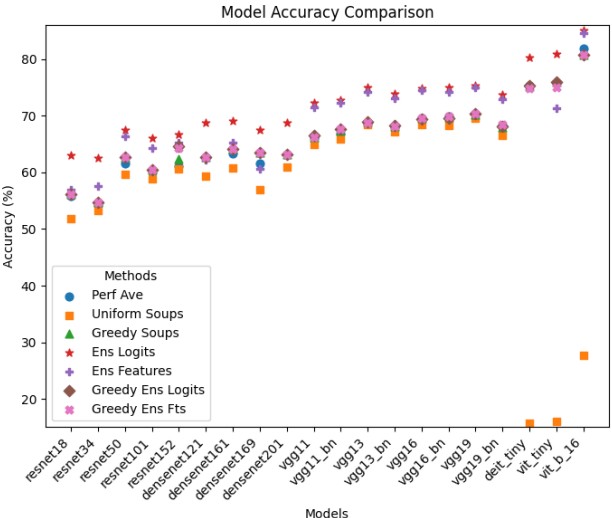

Figure 5: Comparison of model merging and ensembling accuracy across 10 models with different architectures on CIFAR-100. Methods include weight averaging (Uniform Soups, Greedy Soups), feature averaging (Ens Logits, Ens Features, Greedy Ens Logits, Greedy Ens Features), and the average performance of individual models (Performance Average). Experiments are conducted on a range of deep learning architectures, including CNNs and ViTs. Detailed analysis is provided in the experimental section.

In this section, we examine the differences between weight averaging and feature averaging by analyzing their impact and behavior on model effectiveness and efficiency. An illustrative example is provided in Fig. 5.

### 4.1  Linear Model Scenarios

**Averaging on Weights.** Assuming that there is no symmetric neuron mismatch issue and our model only contains linear layers without nonlinearity in between, averaging on weights can be referred to as a process where, given two weight matrices $\mathbf{W}_1$ and $\mathbf{W}_2$, we compute their average first, and then apply the result to the input feature vector $\mathbf{x}$. In our paper, $\overline{\mathbf{W}}$ and $\mathbf{W}$ denote different quantities: $\overline{\mathbf{W}}$ represents the weight tensors after merging, whereas $\mathbf{W}$ refers to the original weights:

$$\overline{\mathbf{h}} = \overline{\mathbf{W}}\mathbf{x} = \frac{1}{2}\left(\mathbf{W}_1 + \mathbf{W}_2\right)\mathbf{x} = \frac{1}{2}\left(\mathbf{W}_1\mathbf{x} + \mathbf{W}_2\mathbf{x}\right). \tag{1}$$

Since both $\mathbf{W}_1$ and $\mathbf{W}_2$ represent linear transformations, the overall transformation remains linear.

**Averaging on features** refers to first applying each weight matrix $\mathbf{W}_1$ and $\mathbf{W}_2$ to the input $\mathbf{x}$ separately, followed by averaging the resulting outputs:

$$\overline{\mathbf{h}}' = \frac{1}{2}\left(\mathbf{h}_1 + \mathbf{h}_2\right) = \frac{1}{2}\left(\mathbf{W}_1\mathbf{x} + \mathbf{W}_2\mathbf{x}\right). \tag{2}$$

In the context of linear operations, the methods in equation 1 and equation 2 are equivalent and yield the same result $\overline{\mathbf{h}} = \overline{\mathbf{h}}'$. That is to say, when no non-linear activation function is involved, averaging on model parameters is mathematically equivalent to averaging on features.

## 4.2 Non-linear Model Scenarios

For weight averaging, the non-linearity is applied after computing the output with the averaged weight matrix:

$$\overline{\mathbf{h}} = \phi\left(\overline{\mathbf{W}}\mathbf{x}\right) = \phi\left[\frac{1}{2}\left(\mathbf{W}_1 + \mathbf{W}_2\right)\mathbf{x}\right] = \phi\left[\frac{1}{2}(\mathbf{W}_1\mathbf{x} + \mathbf{W}_2\mathbf{x})\right]. \tag{3}$$

When averaging on features, the activation function is applied to each output feature independently. The final output is then the average of these activated features:

$$\overline{\mathbf{h}}' = \frac{1}{2}\left(\mathbf{h}_1 + \mathbf{h}_2\right) = \frac{1}{2}\left[\phi\left(\mathbf{W}_1\mathbf{x}\right) + \phi\left(\mathbf{W}_2\mathbf{x}\right)\right]. \tag{4}$$

In this case, the equivalence between $\overline{\mathbf{h}}$ and $\overline{\mathbf{h}}'$ observed in linear models may no longer hold in the presence of non-linearities. For example, with ReLU, if $\mathbf{W}_1\mathbf{x}$ and $\mathbf{W}_2\mathbf{x}$ contain activations of equal magnitude but opposite signs, their average may approach zero. After applying the non-linear activation, this can result in significant information loss. Therefore, averaging in weight space tends to smooth out the contributions from $\mathbf{W}_1$ and $\mathbf{W}_2$, which can be beneficial in some cases by acting as a form of regularization. However, this smoothing may also lead to the loss of fine-grained information. In contrast, feature averaging allows each weight matrix to influence the final output independently after the activation, potentially preserving more detailed structure. However, feature averaging (i.e., model ensembling) increases inference computational cost linearly with the number of models, whereas weight averaging (i.e., model merging) avoids this overhead. The FLOPs for all models used in this study are reported in the supplementary materials.

> **Remark 2**: As shown by the above analyses and the experimental results, feature averaging generally leads to stronger expressive capacity compared to weight averaging. However, weight averaging achieves competitive performance without incurring additional inference costs. This highlights a trade-off between computational efficiency and model performance when choosing between model merging and ensembling, which is an important consideration for aligned and resource-aware AI systems.

# 5 The Model Predictions and Weight Magnitudes

In this section, we examine the behavior of weight-averaged model merging when candidate model weights are uniformly scaled by different constants (e.g., $90\times$, $100\times$), referred to as magnitude factors. This analysis provides insight into the robustness of model merging under varying parameter scales.

## 5.1 Effect of Weight Averaging on Magnitude and Variance of Model Weights

Consider two independent weight matrices $\mathbf{W}_1$ and $\mathbf{W}_2$. Let the averaged weights be defined as

$$\overline{\mathbf{W}} = \frac{1}{2}(\mathbf{W}_1 + \mathbf{W}_2), \tag{5}$$

and let $\|\cdot\|_\infty$ denote the entrywise maximum norm. Define

$$M = \max_{i,j}\left\{|(\mathbf{W}_1)_{i,j}|, |(\mathbf{W}_2)_{i,j}|\right\}. \tag{6}$$

By the triangle inequality Tversky & Gati (1982) under the max norm, we have

$$\|\overline{\mathbf{W}}\|_\infty \leq \frac{1}{2}\left(\|\mathbf{W}_1\|_\infty + \|\mathbf{W}_2\|_\infty\right) \leq M, \tag{7}$$

implying that weight averaging does not increase the maximum entry magnitude.

Assuming element-wise independence between $\mathbf{W}_1$ and $\mathbf{W}_2$, the variance of the averaged weights satisfies

$$\text{Var}(\overline{\mathbf{W}}) = \text{Var}\left(\frac{\mathbf{W}_1 + \mathbf{W}_2}{2}\right) = \frac{1}{4}\left[\text{Var}(\mathbf{W}_1) + \text{Var}(\mathbf{W}_2)\right]. \tag{8}$$

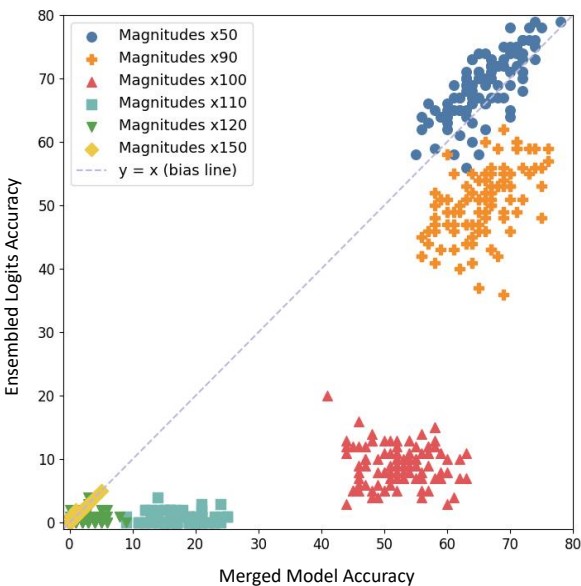

Figure 6: Scatter plot comparing the classification accuracies of model merging (Uniform Soup) and model ensembling (Logits Ensemble) across different weight magnitudes using the VGG19 model on CIFAR-100. Each point represents the accuracy of a sample from a merged or ensembled model, with the X and Y axes indicating the respective accuracies. Detailed analysis is provided in the experimental section.

When both matrices have equal variance $\sigma^2$,

$$\text{Var}(\overline{\mathbf{W}}) = \frac{1}{2}\sigma^2, \tag{9}$$

showing that averaging halves the variance under equal-variance and independence assumptions. More generally,

$$\text{Var}(\overline{\mathbf{W}}) = \frac{1}{4}(\sigma_1^2 + \sigma_2^2) \leq \max(\sigma_1^2, \sigma_2^2), \tag{10}$$

indicating that the averaged model's variance is always less than or equal to the larger of the two original variances.

## 5.2 How Weight Magnitudes and Variance Affects Model Outputs

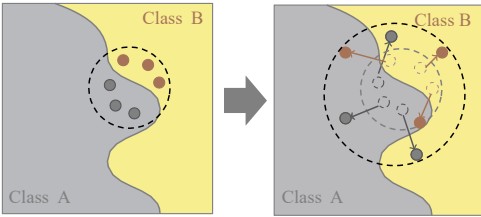

Figure 7: An illustration showing how increasing the magnitudes and variances of model weights leads to larger output magnitudes and variances, causing predictions to shift away from the correct side of the decision boundary.

We have the following property 1 and theorem 1 Wang et al. (2024) to build the connections between model weights' magnitudes/variances and outputs' magnitudes/variances:

**Property 1:** Let $f_\theta(x)$ be a fully-connected neural network, where $\theta$ is the parameters. Assume the activation function $\phi$ is Lipschitz continuous with constant $L$, and the weight matrices $\mathbf{W}^{(m)}$ are random

matrices with independent and identically distributed (i.i.d.) sub-Gaussian entries. Then, for any input $\mathbf{x}$, the output at the $m$-th layer, $\mathbf{y}^{(m)}$, satisfies the following upper bound:

$$\|\mathbf{y}^{(m)}\|_2 \le (L\lambda)^m \|\mathbf{x}\|_2, \text{for } \forall \ \tau > 0 \tag{11}$$

with a probability of at least $\left(1 - 2e^{-\tau^2}\right)^m$, where $(m)$ indexes the neural network layer, with $m$ representing the exponential operation, and $\lambda = \sqrt{N} + C_s K_s^2(\sqrt{N} + \tau)$, with $N$ being the maximum number of neurons across all layers, $C_s$ representing a constant, and $K_s = \max_i \left\|\mathbf{W}_i^{(m)}\right\|_2$. This property shows the output norm can grow exponentially with the number of layers $m$, controlled by the Lipschitz constant of the activation function and an upper bound on the spectral norm of the weight matrices.

**Theorem 1:** Assume the weights in each layer are i.i.d. sub-Gaussian random variables with variance $\sigma_w^2$, and the biases are with variance $\sigma_b^2$. Also, let the activation function $\phi$ be Lipschitz continuous with constant $L$. Then, the variance of the output of the final layer, represented by $\mathbb{V}(f_\theta(x))$, can be bounded by:

$$\begin{aligned}
\mathbb{V}(f_\theta(x)) \le & \|x\|_2^2 \prod_{m=1}^{M_l} [L^2 N (\sigma_w^{(m)})^2] + L^2 N \left(\sigma_b^{(M_l)}\right)^2 + \\
& L^2 \sum_{m=1}^{M-1} \left\{ N \left(\sigma_b^{(m)}\right)^2 \prod_{l=m+1}^{M} \left[ L^2 N \left(\sigma_w^{(m)}\right)^2 \right] \right\},
\end{aligned} \tag{12}$$

where $M_l$ denotes the maximum number of layers, and $L$ represents the Lipschitz constant. This inequality shows that the output variance grows with the depth of the network, and is impacted by the variances of the weights/bias.

Property 1 and Theorem 1 show that the increment of model weights' magnitudes/variances will also magnify outputs' magnitudes/variances. As shown in fig. 7, consider a neural network with input $\mathbf{x}$ and output $\mathbf{y}$, where weights are denoted as $\mathbf{W}$. When model weights exhibit large magnitudes and high variance, predictions become unreliable [1]. These factors can push weights toward unstable values, ultimately harming the model's generalization ability. On the other hand, when $\mathbf{W} \to 0$, the output $\mathbf{y}$ tends toward a fixed value primarily determined by the network's bias terms, becoming largely insensitive to the input $\mathbf{x}$. Hence, if the magnitude of $\mathbf{W}$ is small (but not identically zero), the range of outputs for $\mathbf{y}$ will be constrained. This reduces the sensitivity of $\mathbf{y}$ to the changes of $\mathbf{x}$. From this we can conclude that model-merging exhibits a behavior analogous to L1 or L2 regularization: smaller weight magnitudes suppress excessive responses to minor input perturbations, thereby enhancing model robustness. However, this comes with a trade-off: the reduced expressivity may impair the model's ability to capture complex patterns, potentially degrading performance on challenging tasks.

> **Remark 3**: As analyzed above, model-merging is generally more robust to variations in weight magnitudes and variances compared to model ensembling. This robustness arises because merging tends to reduce the maximum weight magnitudes and variances before inference. Smaller weight magnitudes reduce the model's sensitivity to input perturbations, thereby enhancing stability, but they can also limit the expressiveness gained from ensembling diverse models. From this view, model-merging behaves similarly to regularization. This trade-off between robustness and expressiveness is important and should be evaluated based on task requirements. For tasks where resilience to noise outweighs the need for fine-grained discrimination, smaller weight magnitudes introduced by merging can lead to improved performance.

## 6 Experiments

### 6.1 Experimental Settings

For the **model weight pattern** experiments in the single-layer setting, we train models on CIFAR-100 Krizhevsky (2009) and Tiny ImageNet Le & Yang (2015) using momentum SGD with an initial learning

---

[1]This can result from factors such as task specificity, insufficient regularization, or improper weight initialization.

Table 1: The table shows dataset size, class number, and link for each of the dataset. In particular, ChestXRay dataset has 14 labels and each of them is a binary classification task.

| Datasets | Sizes | Class # | Links |
|---|---|---|---|
| Cifar10 | 60,000 | 10 | https://www.cs.toronto.edu/~kriz/cifar.html |
| CIFAR-100 | 60,000 | 100 | https://www.cs.toronto.edu/~kriz/cifar.html |
| PathMNIST | 107,180 | 9 | https://medmnist.com/ |
| DermaMNIST | 10,015 | 7 | https://medmnist.com/ |
| CelebA | 202,599 | 2 | https://mmlab.ie.cuhk.edu.hk/projects/CelebA.html |
| TinyImageNet | 100,000 | 200 | https://huggingface.co/datasets/zh-plus/tiny-imagenet |
| ChestXRay | 112,120 | 14×2 | https://www.kaggle.com/paultimothymooney/chest-xray-pneumonia |

rate of 0.01, decayed by a factor of 0.1 every 20 epochs. During examination, weights are reshaped to match the input image dimensions (e.g., $100 \times 3 \times 32 \times 32$ for CIFAR-100), as described in the main text. For experiments involving multi-layer deep model (with non-linearity) weight patterns, we train ResNet50 and VGG19 to full convergence.

Regarding **weight averaging versus feature averaging**, we focus on weight-averaged model merging using Uniform Soups and Greedy Soups Wortsman et al. (2022). We design experiments from two perspectives: model-wise and dataset-wise evaluations of model merging and ensembling performance. The considered architectures include ResNet He et al. (2016) (ResNet18–ResNet152), DenseNet Huang et al. (2017) (DenseNet121–DenseNet201), VGG Simonyan (2015) (VGG11–VGG19, with and without batch normalization), ViT Steiner et al. (2022); Dosovitskiy (2021) (Both deit_tiny_patch16_224 and vit_tiny_patch16_224 are Vision Transformer (ViT)–based architectures), and DeiT Touvron et al. (2021). For dataset-wise evaluation, we use CIFAR-10, CIFAR-100, PathMNIST, DermaMNIST Yang et al. (2021; 2023), CelebA Liu et al. (2015), Tiny ImageNet, and ChestXRay14 Wang et al. (2017); Ma et al. (2019). Detailed dataset information is summarized in table 1. Specifically, ChestXRay14 is a multi-label chest X-ray dataset comprising 112,120 frontal images from 30,805 patients, annotated with 14 thoracic disease labels. We follow the official train-test split and report the average Area Under the Receiver Operating Characteristic Curve (AUROC) over all labels as the evaluation metric. In terms of model training, following Wortsman et al. (2022), we train multiple instances of the same network architecture on the dataset, each initialized with a distinct random seed. After training converges, we merge the parameters of a selected subset of these independently trained models.

For model ensembling, we consider ensembling on model logits (pre-softmax or pre-sigmoid outputs) and on intermediate features (pre-classification layer activations). In addition to these two ensembling approaches, whenever a model is selected for inclusion in a Greedy Soup, we also ensemble it with what we refer to as the Greedy Ensemble Logits (Grd Ens Lgt) and Greedy Ensemble Features (Grd Ens Fts) baselines. To avoid bias from adopting the final layer of any individual model, the "Ens Features" approach constructs the classifier by merging the final fully connected (FC) layers from all participating models.

For the experiments on **model predictions and weight magnitudes**, multiple copies of the same architecture are trained independently on the identical dataset with different random initializations; after convergence, parameters (after applying the magnitudes) from chosen models are merged. We use CIFAR-10, CIFAR-100, and Tiny ImageNet. To amplify both the magnitude and variance of model weights, we apply a scaling factor $c$ to all model parameters. Formally, scaling each element of a weight matrix $A$ by $c$ increases the variance to $c^2 \cdot \text{Var}(A)$. For instance, if $c = 100$, the variance becomes $10{,}000 \cdot \text{Var}(A)$.

For all experiments, we follow the official dataset splits to ensure fair comparisons. When a Python interface is publicly available, we use it to directly load the data. Across all settings, we uniformly train 10 models per configuration and perform model merging and ensembling under identical conditions, including learning rates, learning rate schedules, optimizers, and number of models.

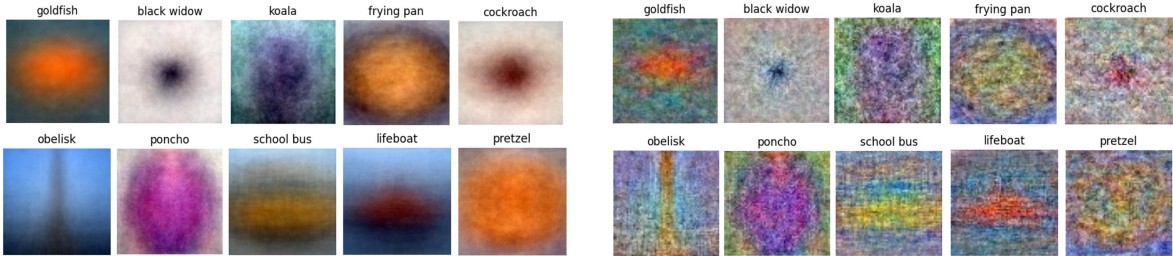

(a) Average of all images for each class.        (b) Linear classifiers visualization.

Figure 8: Average of all images for each class and linear classifiers visualization for each class in Tiny ImageNet Le & Yang (2015) dataset.

## 6.2 The Patterns Contained in Model Weights

### 6.2.1 Class-wise Pattern Examinations

As shown in Fig. 1(a,b), the visualization of the linear classifier on CIFAR-100 reveals class-specific patterns that closely resemble the average image of each class. These learned patterns effectively serve as fused representative templates for their respective categories. Interestingly, if we merge the weights of the first 5 classes with the last 5 classes in fig. 1, it gives us the results of fig. 2. As mentioned, the weight-averaged model-merging acts as a Mixup Zhang (2018) operation, which is a linear combination of different matrices, in the weight space. For instance, when the "bicycle" weight matrix is combined with the "bottle" weight matrix, we can see the wheels of bikes combined with a vertical bottle. We can also observe that, after merging, each element within the combined template noticeably fades. This phenomenon can also explain the limitation of expressiveness of the merged model as described in the experimental section.

Fig. 8 shows the class-wise average images alongside their corresponding linear classifier weights for the Tiny ImageNet dataset. For instance, "goldfish" shows a reddish center in the averaged images and also shows a reddish center in the classifier weights; similarly, we can observe the eyes and nose of "koala"; for "pretzel", we can see the shape of the cracker from its weight visualization. However, the patterns are less clear compared to those in CIFAR-100. This reduced clarity may be due to the larger dataset size, which results in class-wise average images that are themselves more diffuse. Additionally, the limited capacity of a single linear classifier, combined with the increased complexity of visual patterns in Tiny ImageNet, may further obscure the learned representations. In addition to the linear classifier, we also examine the weights of standard deep learning models by visualizing the first-layer convolutional kernels (of size $7 \times 7$) from the ResNet50 model He et al. (2016), as shown in the supplementary material. The examination of the VGG model weights, as well as all class-wise visualizations for CIFAR-100 and Tiny ImageNet, are also included in the supplementary materials.

### 6.2.2 Model Similarity V.S. Accuracy of Merged Models

Fig. 9a illustrates the relationship between model-wise similarity (we measure the similarity between pair of linear models and in total we train 5 models with 5 different seeds) and merged test accuracy. The x-axis represents the mean cosine similarity of the corresponding rows (i.e., templates), measuring how similar different templates are in the embedding space. Higher values indicate greater semantic similarity between templates. The y-axis shows the merged test accuracy (%), reflecting the final performance after merging. Note that cosine similarities are all very close to 1.0; the x-axis uses Matplotlib's offset notation to visualize variations at the $10^{-6}$ scale. Higher template similarity consistently correlates with better merge performance. Each point corresponds to one experimental setting. The solid line denotes a linear regression fit, which reveals an overall positive trend: higher template similarity tends to be associated with higher merged accuracy. To quantify this relationship, we report both Pearson and Spearman correlation coefficients. The Pearson correlation coefficient (r = 0.428) measures the strength of a linear relationship between template similarity and accuracy, while the Spearman rank correlation coefficient ($\rho = 0.390$) assesses

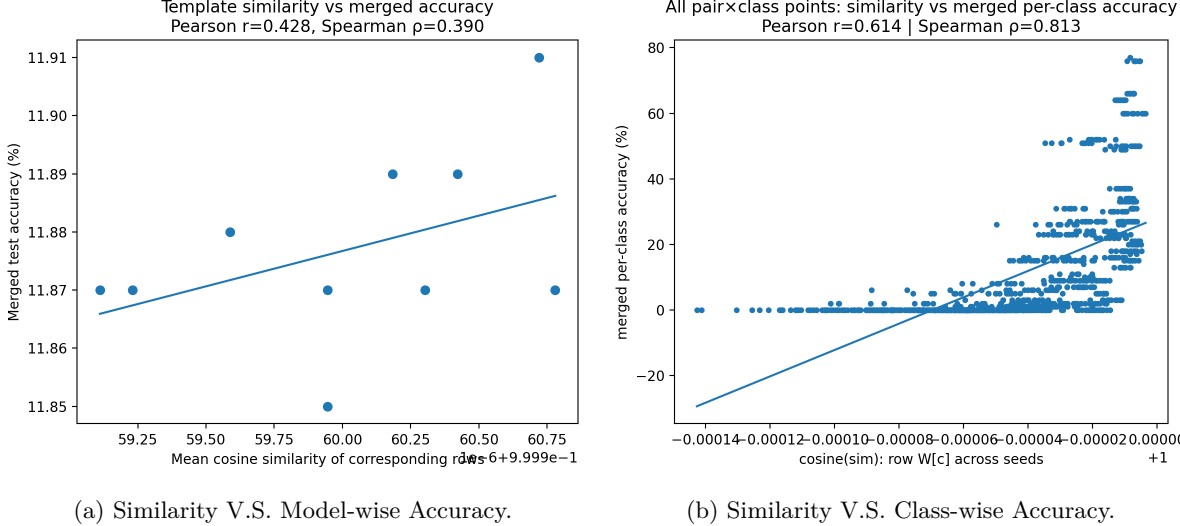

(a) Similarity V.S. Model-wise Accuracy.

(b) Similarity V.S. Class-wise Accuracy.

Figure 9: Similarity V.S. Accuracy of Merged Models.

the monotonic relationship based on ranked values and is less sensitive to outliers. Both metrics indicate a moderate positive correlation, suggesting that increased similarity among templates is generally beneficial for merged performance, although the effect is not strictly linear.

Similarly, class-level relationship between template similarity and merged linear classifier performance (each pair of models has 100 records and in total we train 5 models with 5 different seeds) shown as Fig. 9. Each point corresponds to one *(model pair, class)* instance on CIFAR-100. The x-axis shows the cosine similarity between the corresponding class-specific weight rows of two independently trained linear classifiers, while the y-axis shows the merged model's per-class test accuracy after weight-space averaging. We observe a strong positive correlation between template similarity and merge performance (Pearson $r = 0.614$, Spearman $\rho = 0.813$), indicating that classes with more aligned templates across random seeds tend to merge more successfully. Notably, some class templates tend to suffer from near-zero merged accuracy due to the adoption of just linear classifiers.

### 6.3 Weights vs. Features Averaging

### 6.3.1 Observations over Different Architectures

As shown in Fig. 5 for the merging of 10 models (the experiments for merging from 2 to 7 models are in the supplementary materials). For abbreviations, "Perf Ave" denotes the average performance of individual models; "deit_tiny" and "vit_tiny" denote deit_tiny_patch16_224 and vit_tiny_patch16_224 models, respectively. Different models show different merging/ensembling performances, but model ensemble on logits generally performs the best across different tested architectures He et al. (2016); Huang et al. (2017); Simonyan (2015); Steiner et al. (2022); Dosovitskiy (2021); Touvron et al. (2021).

In contrast, uniform soups perform the worst. Greedy soups generally outperform uniform soups, as they selectively merge models based on individual performance. This strategy ensures that the final result is at least as strong as the best initial model, since poorly performing candidates can be excluded. Greedy ensembling on both logits and features yields performance comparable to that of greedy soups.

In general, incorporating more models into the combination process improves performance across both model-merging and ensembling methods. Notably, Vision Transformer (ViT) architectures exhibit significantly larger performance gaps between uniform weight averaging and logits ensembling, and these gaps widen as more models are added. This may stem from fundamental architectural differences between convolutional neural networks (CNNs) and ViTs. CNNs are inherently suited for learning hierarchical local patterns,

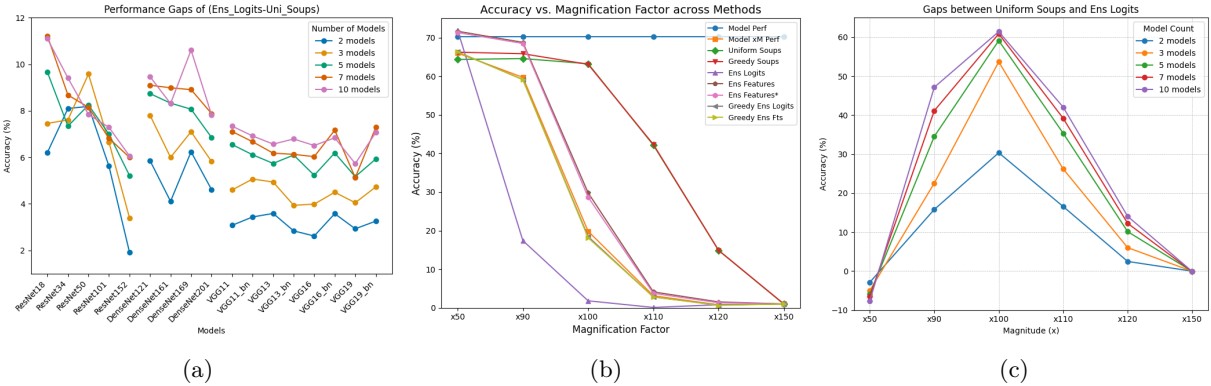

(a)  (b)  (c)

Figure 10: (a) Performance Gaps between Logits Ensemble and Uniform Soups Across Different Configurations (excluded the ViT models as the scales are too different) on CIFAR-100. (b) Accuracy vs. Magnification Factor across Methods with 10 models on CIFAR-100 dataset. (c) Difference in accuracy between Uniform Soup and Logits Ensemble across varying magnification factors and model counts on the CIFAR-100 dataset.

making them more stable under weight-averaged merging. In contrast, ViTs rely on global self-attention mechanisms that capture token-level dependencies, rendering their learned representations more sensitive to perturbations in weights. Similar performance degradation for ViTs is observed on other datasets as well, such as the DeiT model on TinyImageNet (see supplementary materials for details).

We also visualize the gaps between Logits Ensemble and Uniform Soups Across Different Configurations (excluding ViT models as the gaps are too large, so in different scales) in fig. 10(a). We can see that the more models in the merging process, the larger the gap between logits ensemble and uniform soups. The accuracy gaps vary across models but generally decrease as model depth increases (e.g., ResNet18 to ResNet152, DenseNet121 to DenseNet201, VGG11 to VGG19 without BatchNorm). This trend likely reflects performance saturation due to enhanced representational capacity with deeper networks. However, the presence of batch normalization can disrupt this pattern. Additionally, supplementary materials show that greedy soups reduce the gap between logits ensembling and merging, mitigating this marginal effect.

### 6.3.2 Observations over Different Datasets

Table 2: Performance comparison of VGG19 with 10 models across different datasets. For the ChestXRay dataset Wang et al. (2017); Ma et al. (2019), we use the average AUROC (Area Under the Receiver Operating Characteristic Curve), normalized to a 0–100 scale for consistency with other datasets. Accuracy (0%–100%) is used as the evaluation metric for all other datasets.

| Datasets | Perf Ave | Uni Soups | Grd Soups | Ens Lgt | Ens Fts | Grd Ens Lgt | Grd Ens Fts |
|---|---|---|---|---|---|---|---|
| CIFAR-10 | 92.37 | 92.90 | 92.62 | 93.91 | 93.81 | 92.63 | 92.62 |
| CIFAR-100 | 70.34 | 69.63 | 70.25 | 75.36 | 74.96 | 70.39 | 70.32 |
| PathMNIST | 90.04 | 32.42 | 88.48 | 92.95 | 75.17 | 88.59 | 88.34 |
| DermaMNIST | 74.17 | 68.18 | 74.16 | 75.76 | 69.73 | 73.92 | 73.17 |
| Celeba | 92.91 | 53.19 | 93.06 | 93.44 | 93.47 | 93.02 | 93.04 |
| TinyImageNet | 59.90 | 62.31 | 61.95 | 63.28 | 63.53 | 63.53 | 63.58 |
| ChestXRay | 80.66 | 79.29 | 80.43 | 81.77 | 81.07 | 80.43 | 80.43 |

For dataset-wise evaluation, we apply model merging and ensembling across seven widely used datasets of varying sizes, including large-scale datasets such as TinyImageNet and ChestXRay, each containing over 100,000 images. As shown in Tab. 2, ensembling generally outperforms merging. Performance differences across datasets reflect their distinct characteristics. Similar to the model-wise comparison, increasing the number of models merged (results for 2 to 7 models are in the supplementary materials) generally improves performance for both merging and ensembling. Ensembling demonstrates greater stability than merging across different datasets in terms of best and worst performance. For instance, on PathMNIST Yang et al.

(2021; 2023), uniform soups perform poorly (32.42% accuracy), falling far behind ensemble methods, unlike their stronger results on CIFAR-10/100 and Tiny ImageNet. A similar pattern appears on CelebA Liu et al. (2015), where uniform averaging reaches only 53.19%. However, greedy soups alleviate this issue (achieving 88.48% on PathMNIST and 93.06% on CelebA) by effectively selecting the best models. These differences may stem from conflicts among model weights.

### 6.4 Model Predictions and Weight Magnitudes

Increases in weight magnitudes also raise weight variance. As shown in fig. 6 (where each accuracy point for merging and ensembling is computed on the same data batch), for moderate magnitude factors (up to ×50), there is an approximate positive linear correlation between model-merging and ensembling performance. Most points lie above the diagonal bias line, indicating ensembling slightly outperforms merging. However, this trend breaks down at larger magnitude factors ($\geq$ ×90). At ×90, a positive correlation remains, but all points fall below the bias line. Beyond ×100, the correlation disappears entirely. These results demonstrate that model merging exhibits greater robustness to large increases in weight magnitude and variance compared to ensembling.

Moreover, we visualize the merging and ensembling of 10 models (merging results for 2 to 7 models are provided in the supplementary materials) across different magnitude factors, as shown in fig. 10b. In the figure, "Model Perf" denotes the average performance of individual models without weight magnification (hence it remains constant), while "Model $\times M$ Perf" represents performance under different magnification factors. The trends indicate that merged models are more robust than ensembled ones — as the magnitude factor increases, ensembling performance (both logits and features) declines more rapidly. Interestingly, feature ensembling demonstrates greater robustness than logits ensembling across configurations and can even outperform model merging at moderate magnification (e.g., ×50). This suggests feature ensembling is somewhat resistant to changes in weight magnitude. The "Ens Features" method ensembles multiple model features while merging all last fully connected (FC) layers to remove the influence of the FC layer. To further reduce the impact of the merged FC layer, we introduce a variant, "Ens Features*", which directly uses the FC layer from the first model. The similar performance of "Ens Features" and "Ens Features*" indicates low sensitivity to the merging of the final layers.

We further visualize the performance gaps between model merging and ensembling in fig. 10c. As the number of models increases, the gaps generally widen, reaching a peak at a certain magnitude factor (e.g., ×100 for CIFAR-100). Results for merging and ensembling 2 to 10 models across various magnitude factors and datasets, along with the corresponding gaps, are provided in the supplementary materials. Similar patterns are observed on other datasets, though the peak magnitude factors vary.

## 7 Discussion

### 7.1 Implications for Future Model-Averaging Designs

Our empirical analysis reveals several insights that can inform the design of future model-averaging techniques. First, the structured and interpretable weight patterns observed across both CNNs and ViTs imply that averaging procedures may benefit from exploiting such structures. Future methods could incorporate representation- or layer-aligned merging to preserve semantic consistency during averaging. Second, the differing behaviors of weight-space merging and feature-space ensembling suggest that uniform averaging may not be universally optimal. This points toward hybrid or task-adaptive merging schemes that selectively combine elements of both strategies based on model architecture or dataset characteristics. Finally, our investigation into parameter scaling and prediction stability highlights the importance of scale-aware design. Integrating explicit normalization or scale-regularization into merging algorithms may lead to more robust outcomes, particularly for architectures with high sensitivity to parameter perturbations. Together, these observations provide actionable directions for developing more principled and reliable model-averaging frameworks.

## 7.2 Broader Impact

Our findings also highlight potential risks. Vanilla weight averaging can silently degrade performance when models encode conflicting patterns, and this degradation may be concentrated on specific subpopulations or rare inputs. The fragility of ViTs under merging is particularly concerning given their growing adoption in safety-critical systems. We therefore recommend that practitioners to systematically compare merged models against all constituent models.

## 7.3 Limitations

We now clarify that Vision Transformers exhibit higher sensitivity to weight-space merging compared to CNNs, likely due to their reliance on global attention patterns and higher parameter interdependence. While our empirical analysis highlights these differences, a thorough mechanistic understanding remains an open research question. We view this point as a promising future direction.

We also note that our experiments are based on standard classification benchmarks and a selected set of architectures and training regimes. Although these settings are representative, they do not cover all possible model families, task types, or training conditions. Further validation on more diverse tasks (e.g., detection, segmentation), larger-scale backbones, and real-world deployment settings can be pursued in future work.

## 8 Conclusion

In this paper, we systematically analyzed weight-averaged model merging through three novel perspectives. Firstly, we thoroughly examined the weight-averaged model-merging approach through the lens of template matching by visualizing weight patterns across various datasets. Secondly, our comparative study of weight versus feature averaging in merging and ensembling revealed distinct behaviors across architectures and data domains, clarifying when each approach is advantageous. Additionally, we showed that model merging produces more stable predictions than ensembling under variations in parameter magnitudes, contributing to improved robustness and generalization. These results make the behavior of weight-averaged model merging more transparent and interpretable, and we believe they are a useful empirical basis for future theoretical work on its mechanisms. We expect these insights to inspire further research on model-merging techniques and support the development of more interpretable and robust model-averaging methods.

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

## A Theoretical Proofs of Weight Magnitude/Variance Expansion Effects Output Magnitude/Variance

For Eq. 10 in the paper, we aim to prove that:

$$\frac{1}{4}(\sigma_1^2 + \sigma_2^2) \leq \max(\sigma_1^2, \sigma_2^2)$$

Assume without loss of generality that $\sigma_1^2 \leq \sigma_2^2$ (the argument is symmetric if $\sigma_2^2 \leq \sigma_1^2$).

Given that $\max(\sigma_1^2, \sigma_2^2) = \sigma_2^2$, the inequality becomes:

$$\frac{1}{4}(\sigma_1^2 + \sigma_2^2) \leq \sigma_2^2$$

Now, observe the following:

$$\frac{1}{4}(\sigma_1^2 + \sigma_2^2) \le \frac{1}{4}(2\sigma_2^2) = \frac{\sigma_2^2}{2}$$

Since $\frac{\sigma_2^2}{2} \le \sigma_2^2$, we conclude that:

$$\frac{1}{4}(\sigma_1^2 + \sigma_2^2) \le \sigma_2^2 = \max(\sigma_1^2, \sigma_2^2)$$

Thus, the inequality holds true.

For the Property 1 and Theorem 1, according to Wang et al. (2024), we have:

**Lemma 1:** Given a fully-connected network, its weight matrix $\mathbf{W}^{(m)}$ satisfies, for any $\tau > 0$,

$$\left\|\mathbf{W}^{(m)}\right\|_2 \le \sqrt{N} + C_s K_s^2(\sqrt{N} + \tau),$$

with a probability of at least $1 - 2e^{-\tau^2}$, where $C_s$ is a universal constant and $K_s = \max_i \|\mathbf{W}_i^{(m)}\|_2$.

For **Property 1**:

*Proof.* We denote $s_1(A)$ as the maximum singular value of $A$ and set $\lambda = \sqrt{N} + C_s K_s^2(\sqrt{N} + \tau)$ as the upper bound of the maximum singular value of all weight matrices by Lemma 1. The have $\phi(0) = 0$ in Property 1. This assumption holds for commonly used activation functions in our experiments, such as ReLU, GELU, and tanh. Under this assumption, the Lipschitz-based bound follows directly. By Lipschitz property of the activation function $\phi$ and $\|A\mathbf{x}\| = s_1(A)\|\mathbf{x}\|$,

$$\begin{aligned}
\left\|\mathbf{y}^{(m)}\right\|_2 &= \left\|\phi(\mathbf{W}^{(m)}\mathbf{y}_k^{(m-1)})\right\|_2 \\
&= \left\|\phi(\mathbf{W}^{(m)}\mathbf{y}_k^{(m-1)}) - \phi(0)\right\|_2 \\
&\le L\left\|\mathbf{W}^{(m)}\mathbf{y}^{(m-1)} - 0\right\|_2 \\
&= L\left\|\mathbf{W}^{(m)}\mathbf{y}^{(m-1)}\right\|_2 \\
&= Ls_1(\mathbf{W}^{(m)})\left\|\mathbf{y}^{(m-1)}\right\|_2 \\
&\le L\lambda\left\|\mathbf{y}^{(m-1)}\right\|_2 \\
&\le (L\lambda)^2\left\|\mathbf{y}^{(m-2)}\right\|_2 \\
&\qquad \cdots \\
&\le (L\lambda)^m\|\mathbf{x}\|_2.
\end{aligned}$$

Please note that the probability comes from the random-matrix assumption on A (sub-Gaussian weights): we bound $||A||_2$ by a constant with high probability, and then the deterministic inequality implies $||Av||_2 \le sA||v||_2$.

Then we calculate the probability that this inequality holds by Lemma 1,

$$\begin{aligned}
&\mathbb{P}\left(\left\|\mathbf{y}^{(m)}\right\|_2 \le (L\lambda)^m\|\mathbf{x}\|_2\right) \\
&= \prod_{k=1}^{m} \mathbb{P}\left(\left\|\mathbf{W}^{(k)}\mathbf{y}^{(k-1)}\right\|_2 \le s_1(\mathbf{W}^{(k)})\left\|\mathbf{y}^{(k-1)}\right\|_2\right) \\
&\ge (1 - 2e^{-\tau^2})^m.
\end{aligned}$$

$\square$

For **Theorem 1**:

*Proof.* For the first hidden layer, we have a linear transformation for one neuron:

$$\mathbf{W}_i^{(1)}\mathbf{x} + b_i^{(1)}.$$

Given that all weights and biases are i.i.d, we get:

$$\mathbb{V}(\mathbf{W}_i^{(1)}\mathbf{x} + b_i^{(1)}) = \mathbb{V}\left(\sum_{j=1}^{N^{(1)}} \mathbf{W}_{i,j}^{(1)} x_j + b_i^{(1)}\right)$$

$$= \mathbb{V}\left(\sum_{j=1}^{N^{(1)}} \mathbf{W}_{i,j}^{(1)} x_j\right) + \mathbb{V}(b_i^{(1)})$$

$$= \sum_{j=1}^{N^{(1)}} x_j^2 \mathbb{V}(\mathbf{W}_{i,j}^{(1)}) + \mathbb{V}(b_i^{(1)})$$

$$= (\sigma_w^{(1)})^2 \sum_{j=1}^{N^{(1)}} x_j^2 + (\sigma_b^{(1)})^2$$

$$= (\sigma_w^{(1)})^2 \|\mathbf{x}\|_2^2 + (\sigma_b^{(1)})^2.$$

By the Lipschitz property of the activation function, we have:

$$f_i^{(1)} = \phi(\mathbf{W}_i^{(1)}\mathbf{x} + b_i^{(1)}) \le L(\mathbf{W}_i^{(1)}\mathbf{x} + b_i^{(1)}).$$

Then,

$$\mathbb{V}(f_i^{(1)}) \le \mathbb{V}[L(\mathbf{W}_i^{(1)}\mathbf{x} + b_i^{(1)})]$$

$$\le L^2 \mathbb{V}(\mathbf{W}_i^{(1)}\mathbf{x} + b_i^{(1)})$$

$$\le L^2 \left[(\sigma_w^{(1)})^2 \|\mathbf{x}\|_2^2 + (\sigma_b^{(1)})^2\right].$$

Hence,

$$\mathbb{V}(f^{(1)}) = \sum_{j=1}^{N} \mathbb{V}(f_i^{(1)})$$

$$\le L^2 N \left[(\sigma_w^{(1)})^2 \|\mathbf{x}\|_2^2 + (\sigma_b^{(1)})^2\right].$$

Similarly, for other layers:

$$\mathbb{V}(f^{(m)}) \le L^2 N^{(m-1)} \left[(\sigma_w^{(m)})^2 \|f'^{(m-1)}\|_2^2 + (\sigma_b^{(m)})^2\right].$$

This result follows by iteratively applying the inequalities, concluding the proof. □

## B More Visualization of Model Weight Patterns

### B.1 Visualize Patterns Captured in Weights of 100 Classes for CIFAR-100 Dataset

Following the visualization approach in the main paper, we present the per-class average images fig. 11 and the corresponding linear classifiers fig. 12 for all 100 classes in CIFAR-100. A clear correlation is observed between the average image of each class and its learned weight pattern.

We further visualize the linear classifiers obtained from pairwise merging between the first 50 and the last 50 classes in CIFAR-100, as shown in fig. 13. The resulting weight patterns exhibit compositional characteristics, such as combinations of class templates (e.g., "motorcycle + woman").

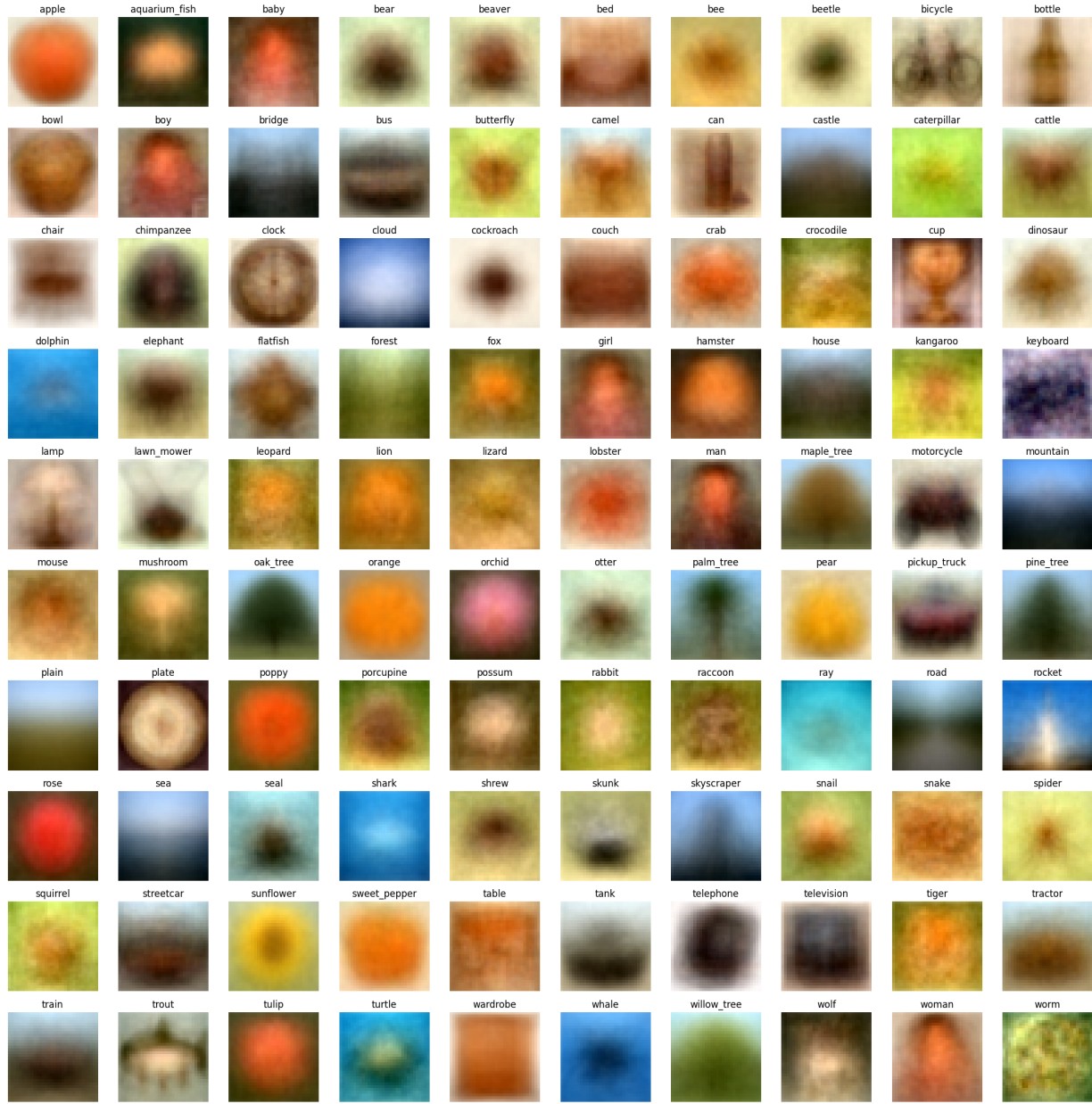

Figure 11: Per-class average images for all 100 classes of the CIFAR-100 dataset.

## B.2 Visualization of Deep Convolutional Kernels Trained on CIFAR-100 Dataset

Additional results of convolutional kernels on CIFAR-100 are shown for deep models, including ResNet50 fig. 14 and VGG19 fig. 15. In ResNet50, we observe both similarities and differences among learned kernel patterns. For example, Kernel 5 and Kernel 9 exhibit similar textures and grayscale distributions, likely capturing similar features and thus producing similar activations (e.g., inner products) with input patches. In contrast, Kernel 9 and Kernel 45 differ substantially. Compared to VGG19, the kernels in ResNet50 are more interpretable, due to their larger spatial dimensions.

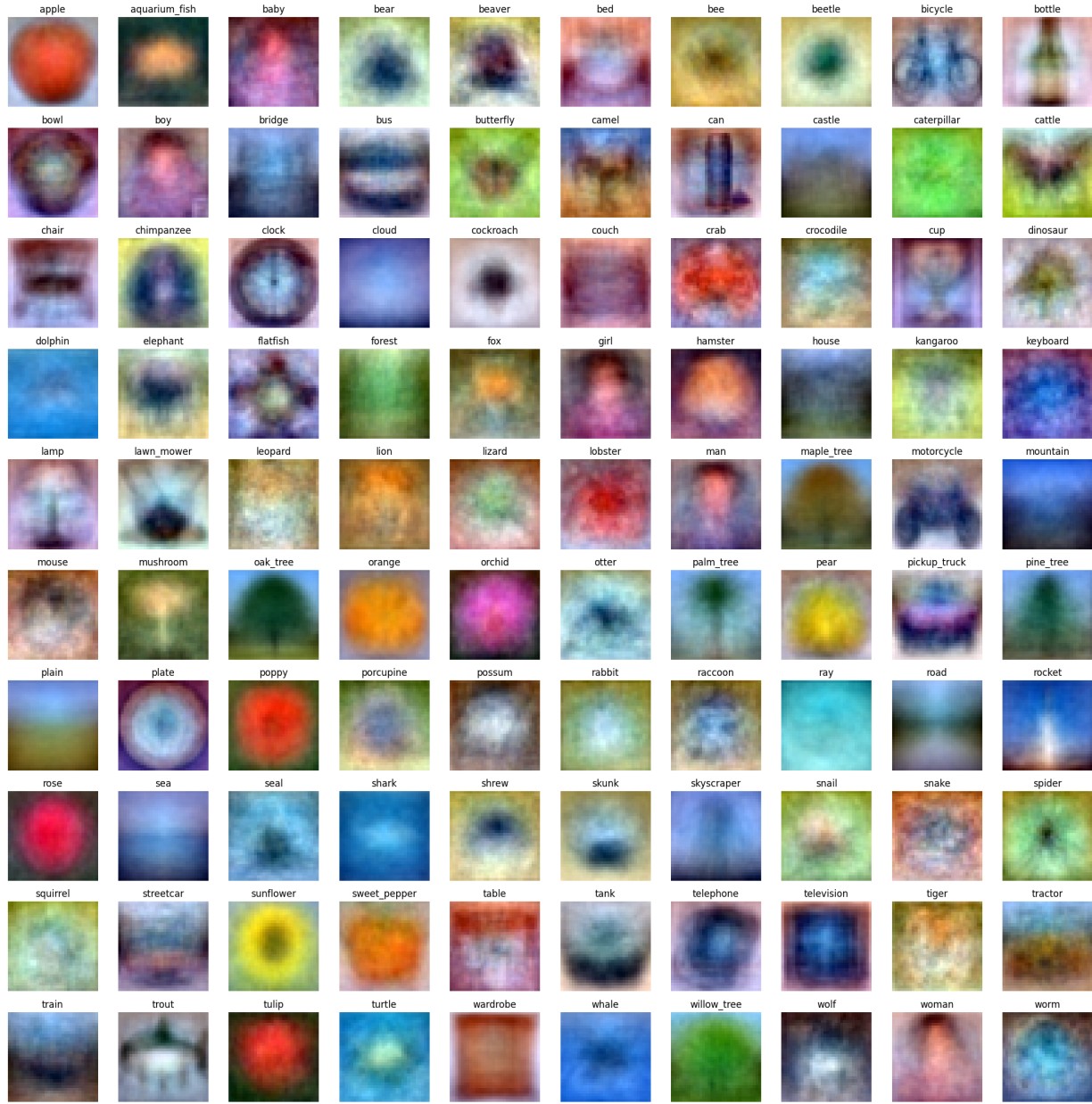

Figure 12: Linear classifiers visualization for each class of CIFAR-100 dataset (all 100 classes).

## B.3 Visualize Patterns Captured in Weights of 200 Classes for TinyImageNet Dataset

For Tiny ImageNet, we present the per-class average images in figs. 16 and 18 and the corresponding linear classifier visualizations in figs. 17 and 19, covering all 200 classes. Similar to CIFAR-100, we observe strong correlations between the average images and the learned classifier weights.

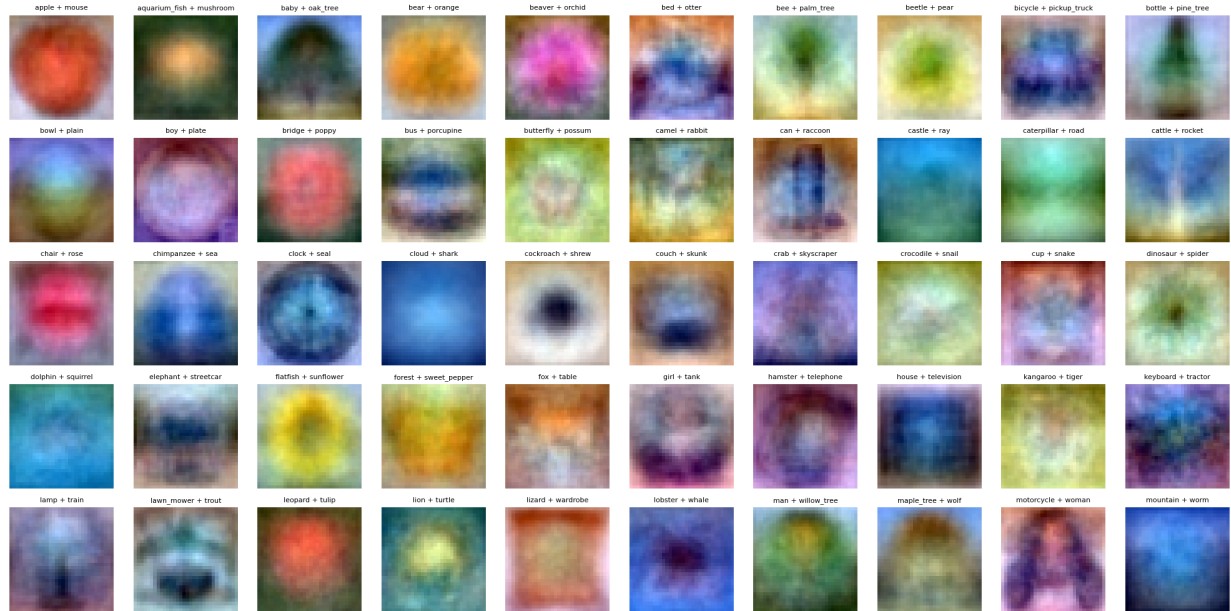

Figure 13: Merge linear classifiers of the first 50 and last 50 classes in the CIFAR-100 dataset.

Table 3: Performance comparison of task merging across different datasets using various model averaging methods. Model1 refers to a VGG19 model trained on CIFAR-100, achieving a classification accuracy of 70.37%. Model2 denotes a VGG19 model trained on a different dataset but evaluated on CIFAR-100. For example, a single Model2 trained on CIFAR-10 yields 5.84% accuracy on CIFAR-100; however after Uniform Souping, the accuracy is improved to 49.63%.

| Datasets | Data Size | Model2 | Uniform Soups | Ens Lgt | Ens Features |
|----------|-----------|--------|---------------|---------|--------------|
| CIFAR-10 | 60,000 | 5.84 | 49.63 | 69.93 | 69.93 |
| PathMNIST | 107,180 | 1.29 | 24.25 | 69.88 | 70.00 |
| DermaMNIST | 10,015 | 5.77 | 60.37 | 69.06 | 69.36 |
| Celeba | 202,599 | 1.03 | 49.33 | 69.48 | 69.61 |
| TinyImageNet | 100,000 | 18.16 | 61.02 | 58.40 | 58.52 |
| ChestXRay | 112,120 | 15.18 | 61.38 | 68.18 | 68.33 |

# C    More Experimental Results of Averaging on Weights V.S. Averaging on Features

## C.1    Merging Models Trained over Different Tasks

We investigate model merging between well-trained models on different tasks, summarized in table 3. Model1 is a VGG19 trained on CIFAR-100 with 70.37% accuracy, while Model2 refers to a VGG19 trained on other datasets but evaluated on CIFAR-100 or after merging or ensembling. Uniform souping using Model2 trained on Tiny ImageNet achieves the highest accuracy (61.02%), outperforming logit ensembling (58.40%). This likely reflects Tiny ImageNet's larger size and diversity, encouraging Model2 to learn more varied features. Although CIFAR-10 is similar to CIFAR-100, souping does not outperform ensembling, possibly due to its small scale cannot contribute much to model learned features. In contrast, ChestXRay, which differs substantially from CIFAR-100, shows notable improvement with souping over Model2 alone, indicating dataset size also influences merging effectiveness. Souping with CelebA-trained Model2 shows mediocre performance, likely because CelebA contains only human faces, which differ largely from CIFAR-100 classes.

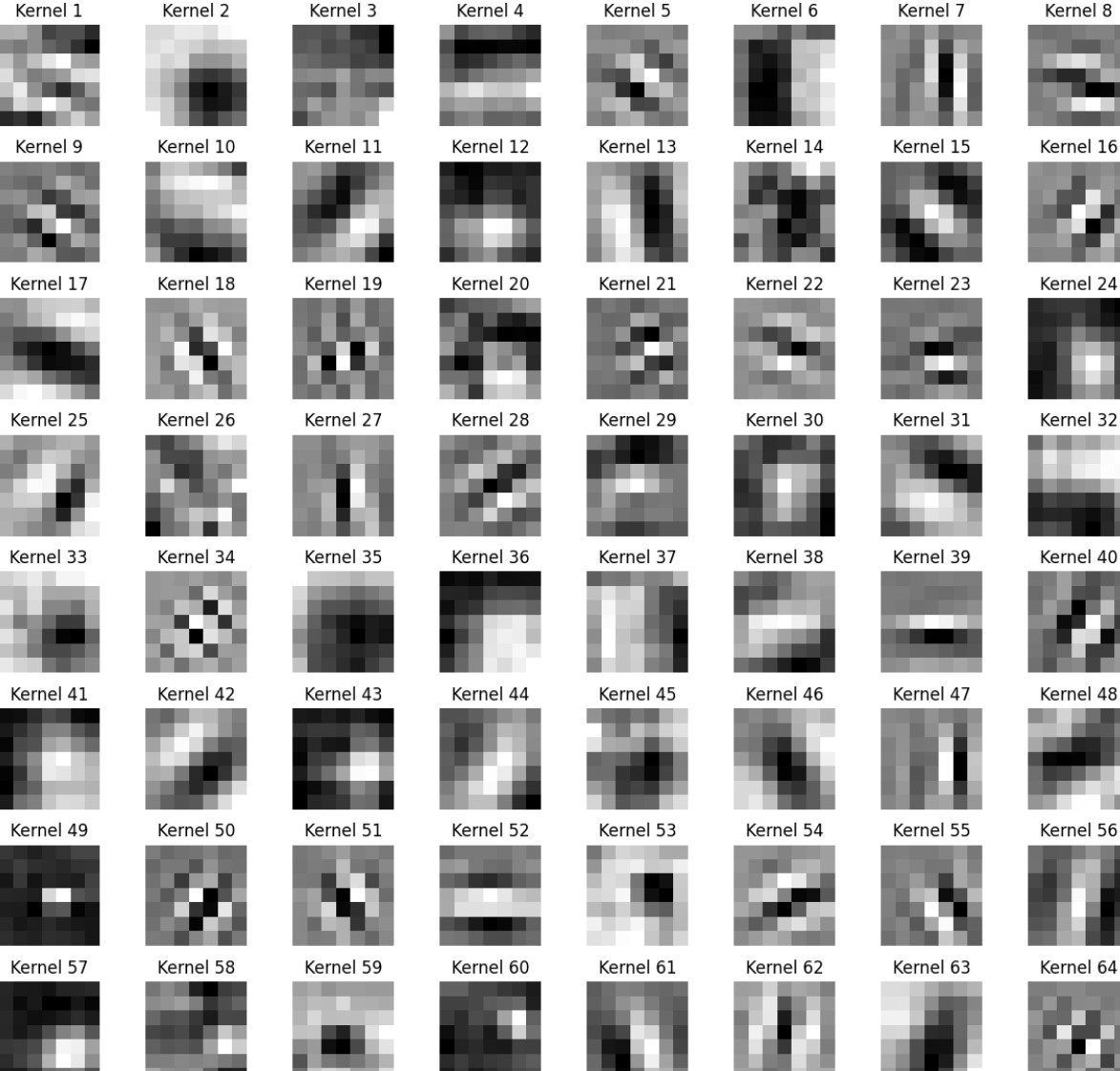

Figure 14: Kernel visualization of ResNet50.

## C.2  2 Models, 3 Models, 5 Models and 7 Models for Model Merging/Ensembling Accuracy Comparison

Model merging and ensembling results using 2, 3, 5, and 7 models across different architectures on CIFAR-100 are shown in figs. 20a, 20b, 21a and 21b, respectively. Across all settings, ensembling on logits consistently yields the highest accuracy, while uniform soups perform the worst. Additionally, model merging is notably less effective for ViT architectures, and the performance gap between uniform soups and logit ensembling widens as more models are included.

Fig. 22 shows the performance gap between ensembling on logits and greedy soups. Similar to the case of uniform soups, the gap generally increases as more models are merged. However, unlike uniform soups,

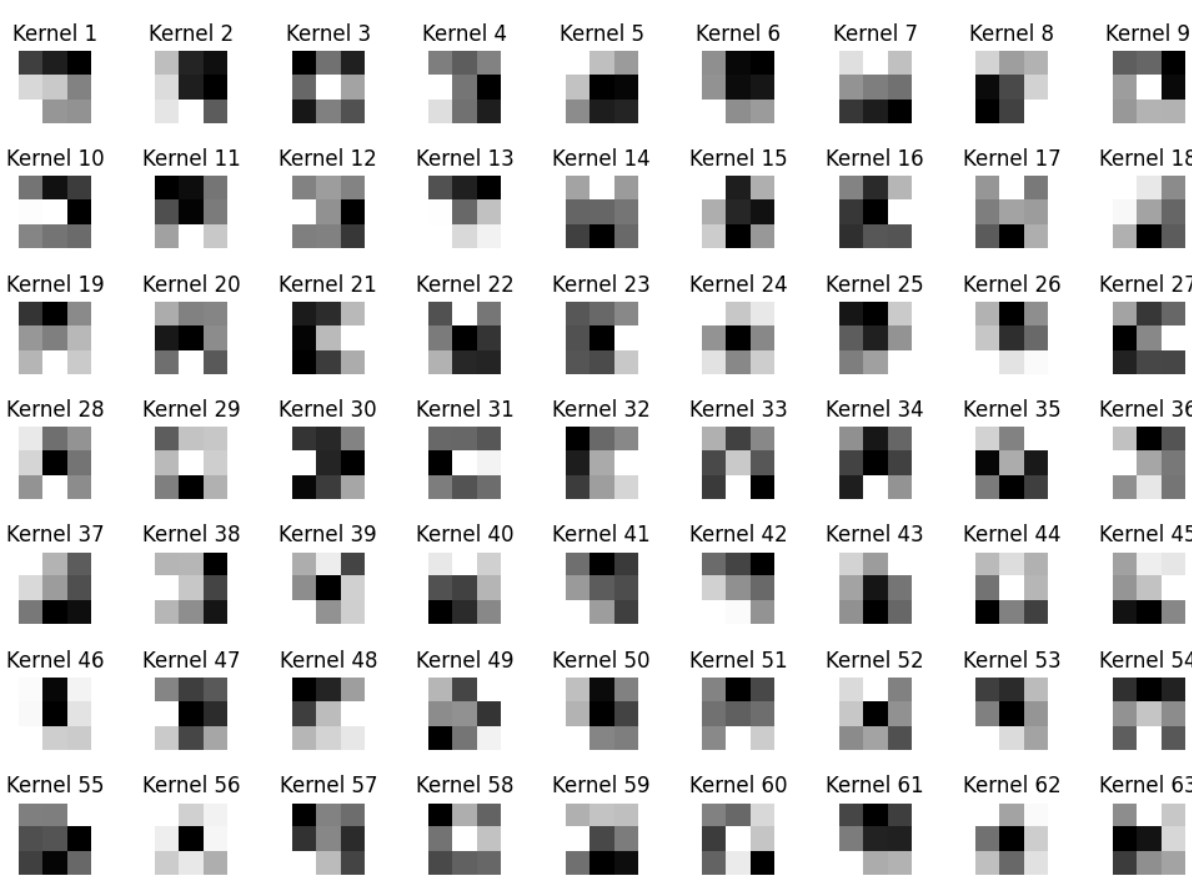

Figure 15: Kernel visualization of VGG19.

greedy soups avoid a consistent performance drop, as they retain relatively high accuracy by only including the first (best) model in the merge process.

Table 4: Performance comparison of VGG19 using 2-model merging across different datasets. The "Perf Ave" denotes the average performance per individual model. For the ChestXRay dataset, we use the average AUROC as the evaluation metric, normalized to a [0, 100] scale for consistency. For all other datasets, classification accuracy (0%–100%) is used.

| Datasets | Perf Ave | Uni Soups | Grd Soups | Ens Lgt | Ens Fts | Grd Ens Lgt | Grd Ens Fts |
|---|---|---|---|---|---|---|---|
| CIFAR-10 | 92.42 | 92.79 | 92.68 | 93.26 | 93.22 | 92.68 | 92.59 |
| CIFAR-100 | 70.31 | 70.06 | 70.22 | 72.99 | 72.76 | 70.57 | 70.41 |
| PathMNIST | 89.75 | 40.85 | 88.41 | 91.60 | 82.06 | 88.61 | 88.47 |
| DermaMNIST | 74.19 | 71.97 | 73.57 | 74.66 | 72.72 | 73.87 | 73.57 |
| Celeba | 93.12 | 89.50 | 93.05 | 93.28 | 93.26 | 93.08 | 93.06 |
| TinyImageNet | 59.87 | 60.89 | 61.40 | 61.71 | 61.85 | 61.72 | 61.65 |
| ChestXRay | 80.52 | 79.80 | 80.43 | 81.14 | 80.85 | 80.43 | 80.43 |

We also present dataset-wise evaluations of 2-, 3-, 5-, and 7-model merging/ensembling in tables 4 to 7, respectively. The corresponding computational costs, measured in Floating Point Operations (FLOPs) and parameter counts, are summarized in table 8.

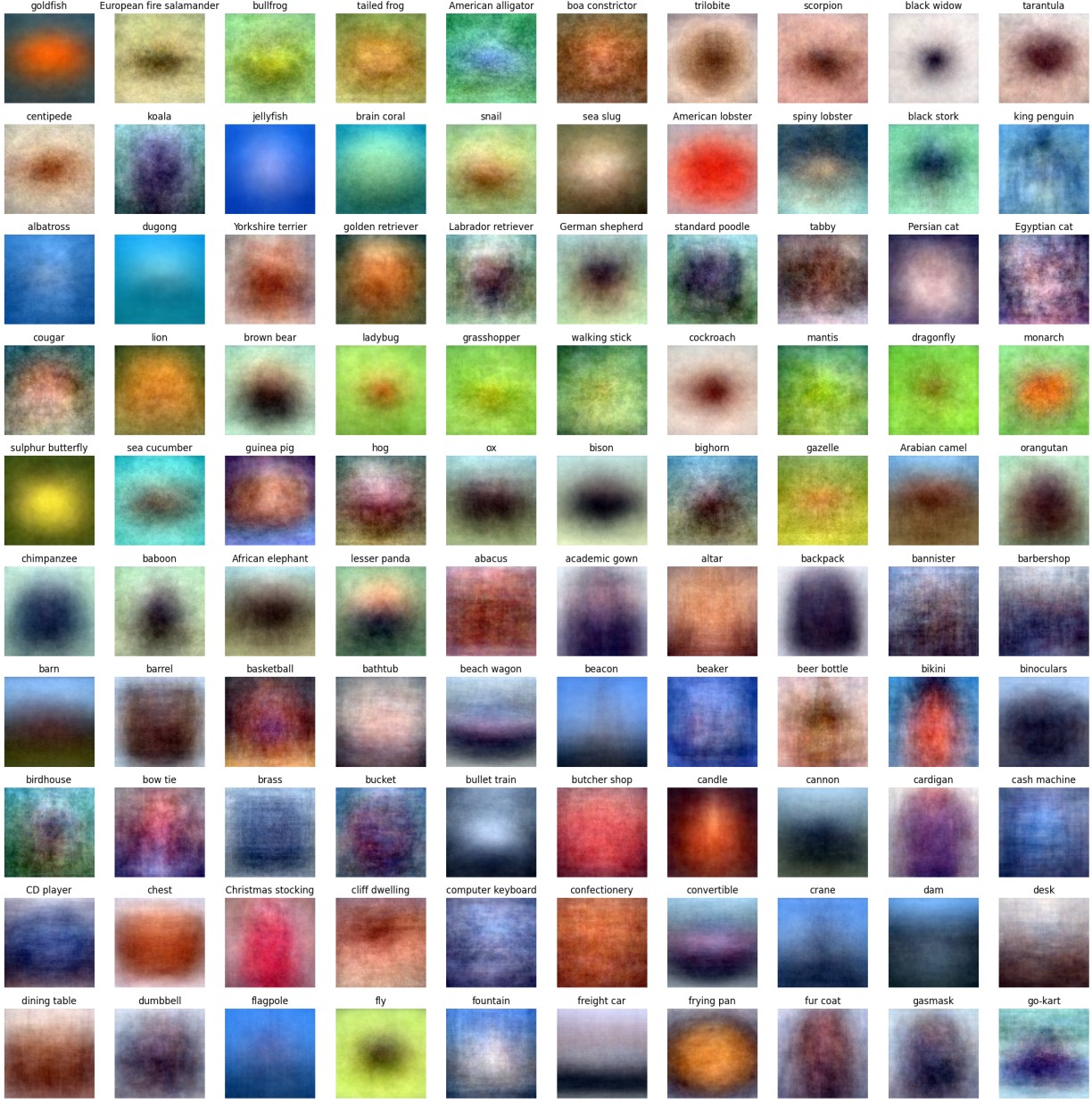

Figure 16: Per-class average images for the first 100 classes of the TinyImageNet dataset.

Table 5: Performance comparison of VGG19 using 3-model merging across different datasets. The "Perf Ave" denotes the average performance per individual model.

| Datasets | Perf Ave | Uni Soups | Grd Soups | Ens Logits | Ens Fts | Grd Ens Logits | Grd Ens Fts |
|---|---|---|---|---|---|---|---|
| CIFAR-10 | 92.47 | 92.82 | 92.65 | 93.54 | 93.48 | 92.75 | 92.65 |
| CIFAR-100 | 70.34 | 69.80 | 70.14 | 73.85 | 73.49 | 70.29 | 70.40 |
| PathMNIST | 89.46 | 45.39 | 88.65 | 91.82 | 80.07 | 88.41 | 88.38 |
| DermaMNIST | 75.00 | 69.43 | 74.01 | 75.21 | 72.07 | 73.47 | 73.37 |
| Celeba | 93.12 | 77.69 | 93.12 | 93.31 | 93.30 | 93.01 | 93.06 |
| TinyImageNet | 59.85 | 61.39 | 61.55 | 62.40 | 62.41 | 62.60 | 62.61 |
| ChestXRay | 80.56 | 79.59 | 80.43 | 81.31 | 80.88 | 80.43 | 80.43 |

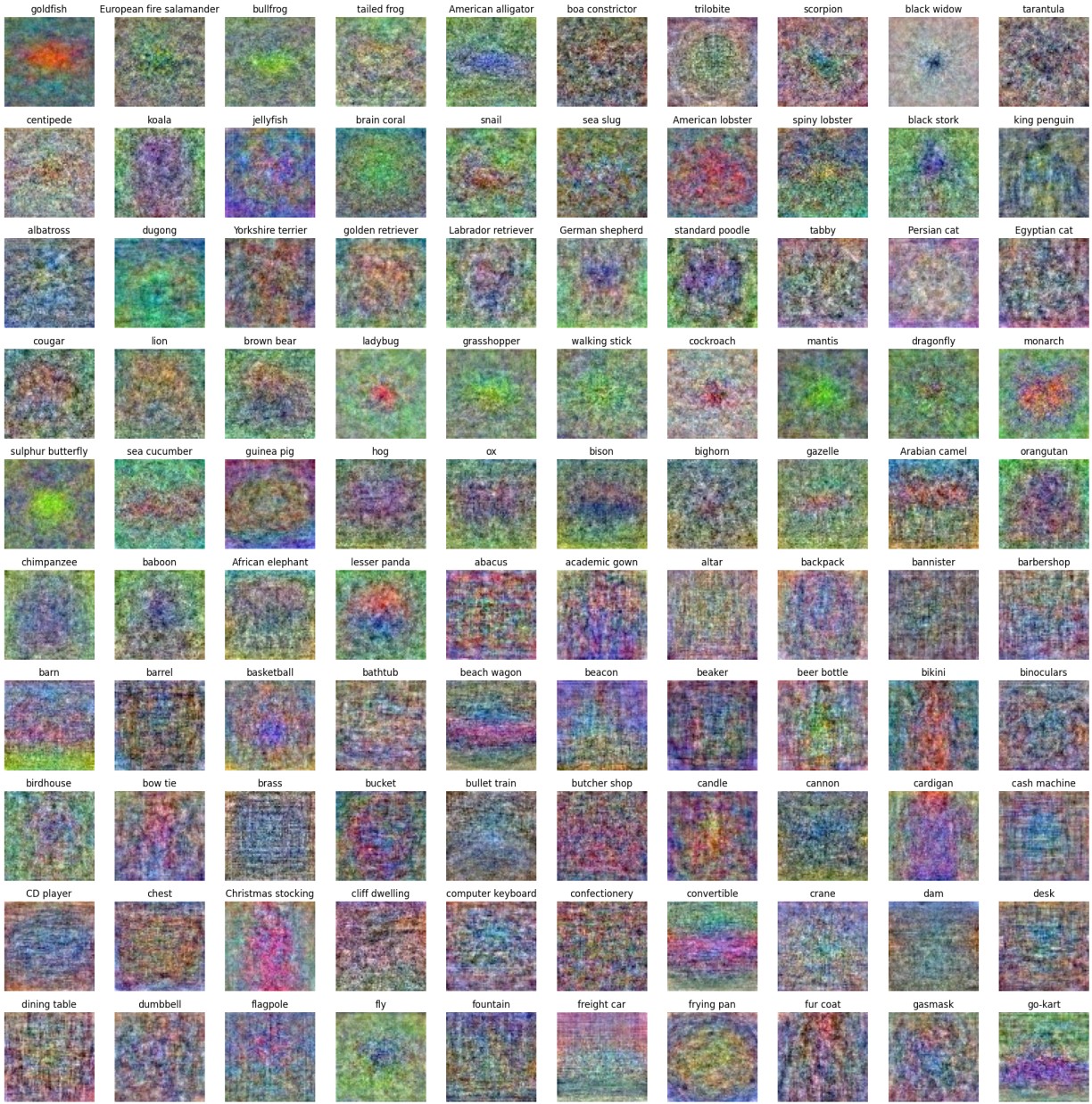

Figure 17: Per class linear classifier visualization in TinyImageNet dataset (the first 100 classes).

Table 6: Performance comparison of VGG19 using 5-model merging across different datasets. The "Perf Ave" denotes the average performance per individual model.

| Datasets | Perf Ave | Uni Soups | Grd Soups | Ens Logits | Ens Fts | Grd Ens Logits | Grd Ens Fts |
|---|---|---|---|---|---|---|---|
| CIFAR-10 | 92.45 | 92.90 | 92.65 | 93.70 | 93.71 | 81.95 | 92.65 |
| CIFAR-100 | 70.47 | 69.73 | 70.32 | 74.90 | 74.18 | 70.35 | 70.31 |
| PathMNIST | 89.80 | 51.00 | 88.58 | 92.28 | 79.42 | 88.47 | 88.41 |
| DermaMNIST | 74.46 | 69.88 | 73.32 | 75.31 | 72.42 | 73.57 | 73.72 |
| Celeba | 93.03 | 60.27 | 93.06 | 93.45 | 93.32 | 93.06 | 93.04 |
| TinyImageNet | 59.89 | 61.55 | 61.91 | 63.08 | 63.17 | 63.29 | 63.27 |
| ChestXRay | 80.57 | 79.38 | 80.43 | 81.46 | 80.92 | 80.43 | 80.43 |

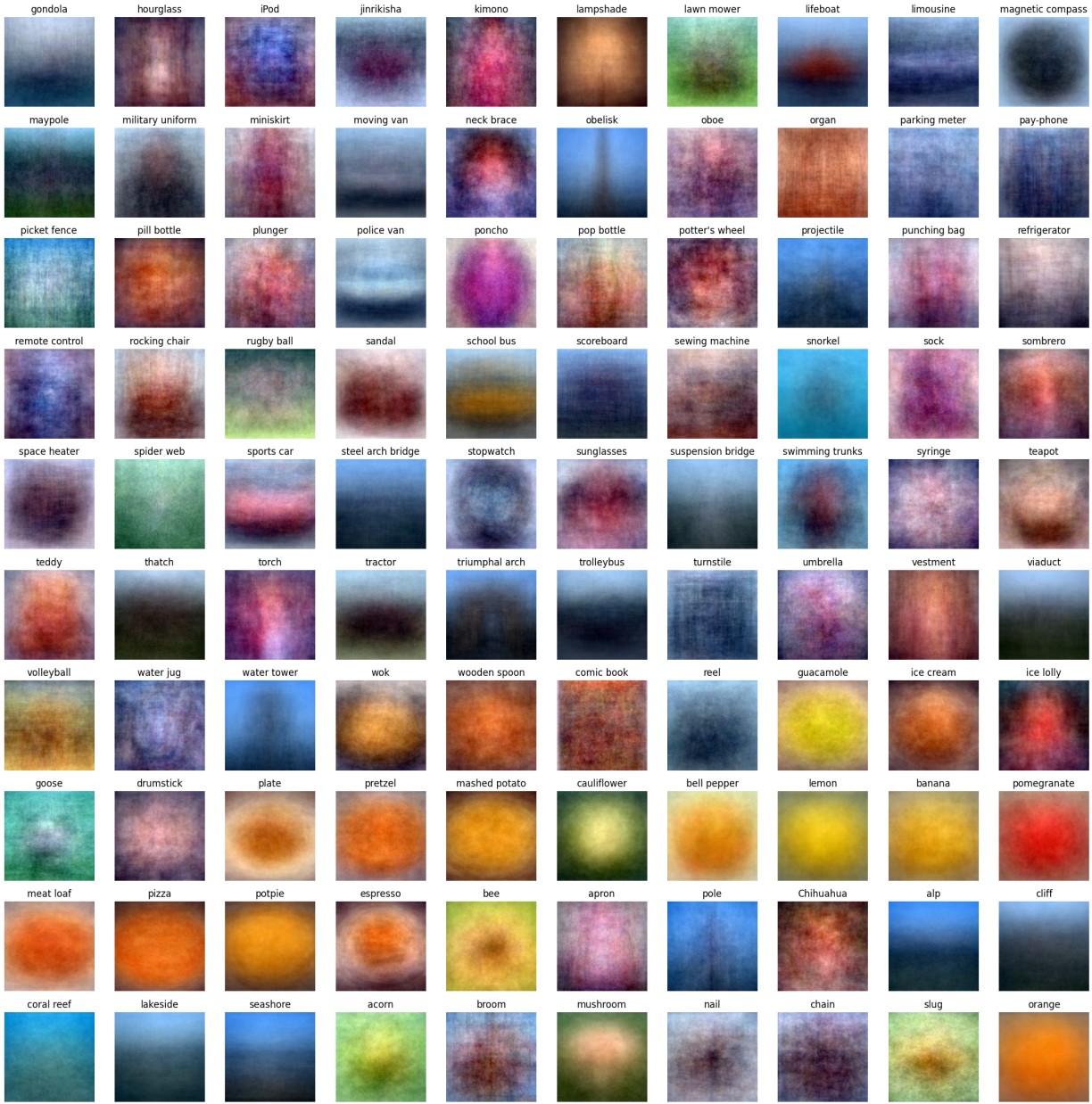

Figure 18: Per-class average images for the last 100 classes of the TinyImageNet dataset.

Table 7: Performance comparison of VGG19 using 7-model merging across different datasets. The "Perf Ave" denotes the average performance per individual model.

| Datasets | Perf Ave | Uni Soups | Grd Soups | Ens Logits | Ens Fts | Grd Ens Logits | Grd Ens Fts |
|---|---|---|---|---|---|---|---|
| CIFAR-10 | 92.42 | 92.79 | 92.65 | 93.85 | 93.71 | 92.76 | 92.71 |
| CIFAR-100 | 70.35 | 69.79 | 70.28 | 74.92 | 74.82 | 70.31 | 70.40 |
| PathMNIST | 89.66 | 38.76 | 88.44 | 92.48 | 79.97 | 88.50 | 88.54 |
| DermaMNIST | 74.12 | 69.83 | 73.67 | 75.86 | 70.57 | 73.82 | 74.06 |
| Celeba | 92.99 | 49.97 | 93.07 | 93.31 | 93.30 | 93.05 | 93.05 |
| TinyImageNet | 59.90 | 62.38 | 62.12 | 63.53 | 63.54 | 63.66 | 63.52 |
| ChestXRay | 80.60 | 79.22 | 80.43 | 81.61 | 80.99 | 80.43 | 80.43 |

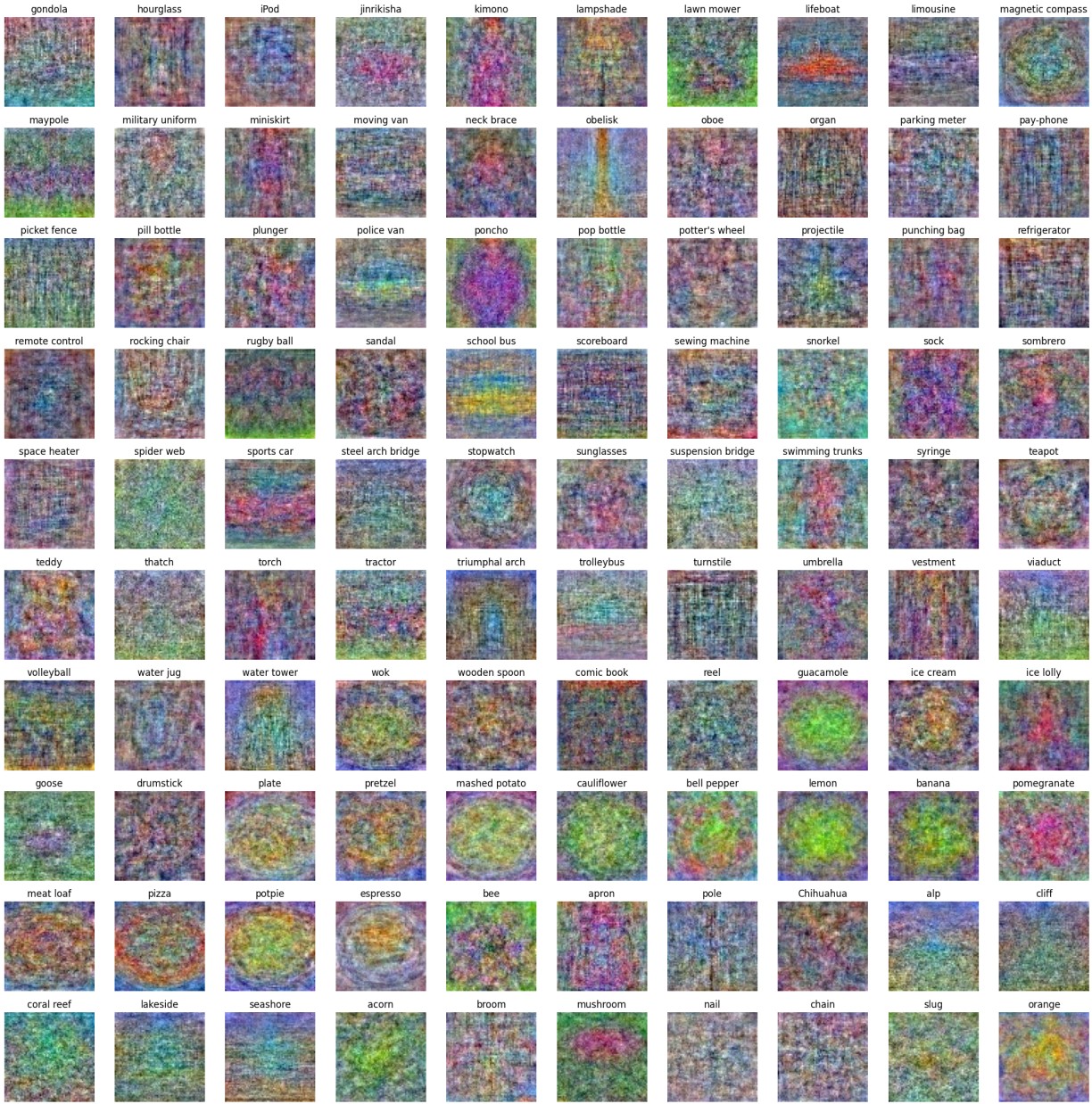

Figure 19: Per class linear classifier visualization in TinyImageNet dataset (the last 100 classes).

We evaluate the ViT model (DeiT-Tiny) on the Tiny ImageNet dataset, as shown in table 9. As more models are merged, the performance of uniform soups degrades further. In contrast, the results of greedy soups, greedy ensembling on logits, and greedy ensembling on features remain unchanged. This indicates that merging two or more ViT models does not improve performance, suggesting that ViT models are not well-suited for model merging in the tested settings.

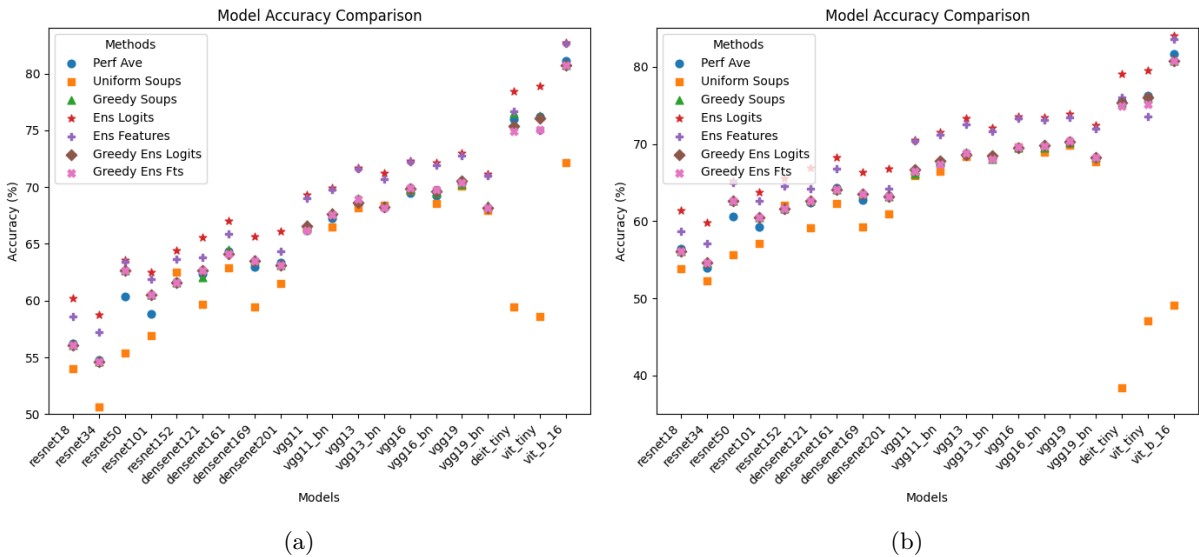

Figure 20: (a) Accuracy comparison of 2-model merging and ensembling across different architectures on CIFAR-100. (b) Accuracy comparison of 3-model merging and ensembling across different architectures on CIFAR-100.

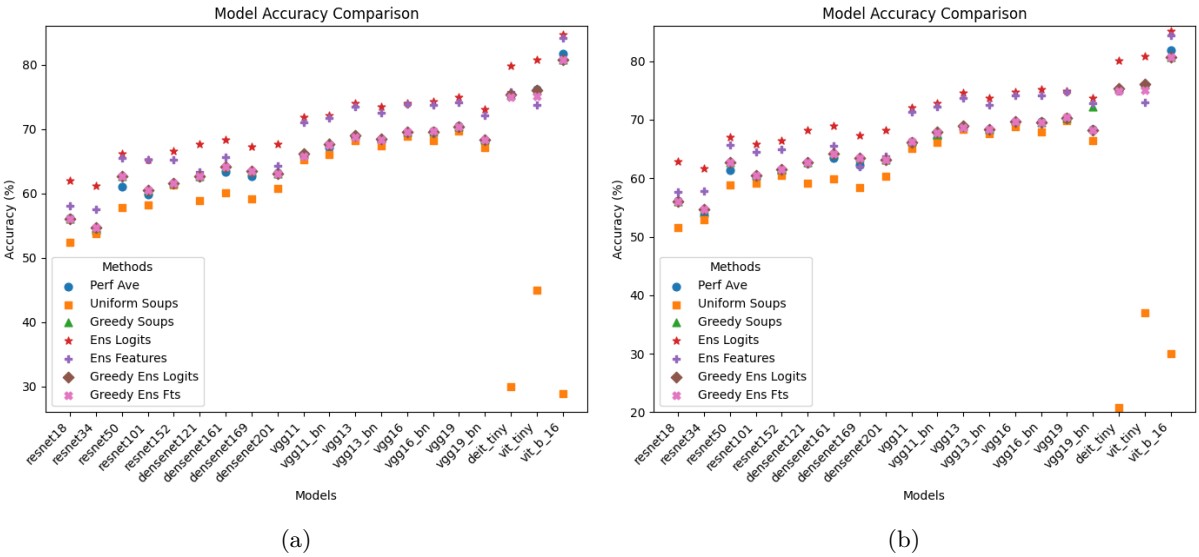

Figure 21: (a) Accuracy comparison of 5-model merging and ensembling across different architectures on CIFAR-100. (b) Accuracy comparison of 7-model merging and ensembling across different architectures on CIFAR-100.

# D  More Experimental Results of Model Magnitude Changes

## D.1  Model-merging can mitigate weight magnitude issue on Tiny ImageNet

We further conduct the weight magnitude experiment on the larger and more complex Tiny ImageNet dataset, as shown in Tab. 10. The results indicate that model merging can still mitigate the effects of weight magnification and, in some cases, outperform logit ensembling, though it remains inferior to feature ensembling. Overall, the effectiveness of model merging on Tiny ImageNet is lower than on CIFAR-100. This may be due to the increased dataset complexity. As discussed in the main paper, model merging can

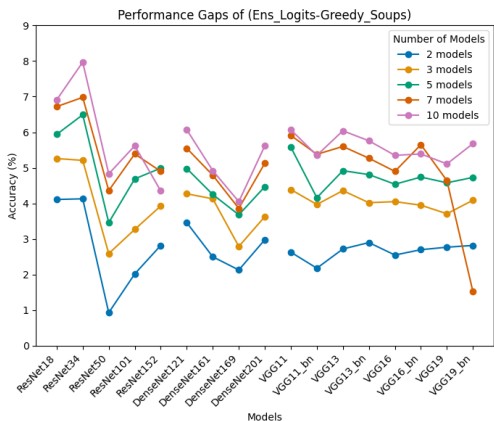

Figure 22: Performance Gaps between Logits Ensemble and Greedy Soups Across Different Configurations (excluded the ViT models as the scales are too different) on CIFAR-100.

Table 8: Comparison of FLOPs and parameter counts for various models. FLOPs are computed using an input resolution of 224 × 224, consistent with standard settings such as ImageNet. "ViT_Tiny" refers to "ViT_Tiny_Patch16_224", and "DeiT_Tiny" refers to "DeiT_Tiny_Patch16_224".

| Models | FLOPs (G) | Param # (M) |
|---|---|---|
| ResNet18 | 1.82 | 11.7 |
| ResNet34 | 3.66 | 21.8 |
| ResNet50 | 4.09 | 25.6 |
| ResNet101 | 7.83 | 44.5 |
| ResNet152 | 11.58 | 60.2 |
| DenseNet121 | 2.88 | 8.0 |
| DenseNet161 | 7.81 | 28.7 |
| DenseNet169 | 3.36 | 14.3 |
| DenseNet201 | 4.37 | 20.0 |
| VGG11 | 7.63 | 132.9 |
| VGG11_BN | 7.76 | 132.9 |
| VGG13 | 11.34 | 133.0 |
| VGG13_BN | 11.47 | 133.0 |
| VGG16 | 15.5 | 138.4 |
| VGG16_BN | 15.52 | 138.4 |
| VGG19 | 19.6 | 143.7 |
| VGG19_BN | 19.63 | 143.7 |
| DeiT_Tiny | 1.3 | 5.7 |
| ViT_Tiny | 1.3 | 5.7 |
| ViT_B_16 | 17.6 | 86.4 |

Table 9: Performance Comparison for DeiT on Tiny ImageNet data across different Configurations. We adopt accuracy as the evaluation metric.

| Model # | Perf Ave | Uniform Soups | Greedy Soups | Ens Logits | Ens Features | Greedy Ens Logits | Greedy Ens Fts |
|---|---|---|---|---|---|---|---|
| 2 models | 66.77 | 61.75 | 65.99 | 70.32 | 69.37 | 65.99 | 65.77 |
| 3 models | 66.32 | 29.93 | 65.99 | 71.56 | 70.31 | 65.99 | 65.77 |
| 5 models | 66.08 | 18.71 | 65.99 | 72.49 | 70.86 | 65.99 | 65.77 |
| 7 models | 66.84 | 13.95 | 65.99 | 73.32 | 72.05 | 65.99 | 65.77 |
| 10 models | 66.79 | 8.91 | 65.99 | 73.93 | 72.58 | 65.99 | 65.77 |

act as a form of implicit regularization, helping to control weight magnitude growth at the cost of reduced model expressiveness.

Table 10: Model merging for mitigating weight magnitude issues is evaluated on Tiny ImageNet using VGG19 models with a magnitude factor of ×100 and accuracy as the evaluation metric. "Org Perf Ave" denotes the model averaged accuracy before weight magnification. While this approach is effective on smaller datasets such as CIFAR-10 and CIFAR-100 due to its regularization effect, it is less effective on larger datasets like Tiny ImageNet, where the loss in model expressiveness outweighs the regularization benefits.

| Model # | Org Perf Ave | xM Perf | Uni Soups | Grd Soups | Ens Lgt | Ens Fts | Ens Fts* | Grd Ens Lgt | Grd Ens Fts |
|---|---|---|---|---|---|---|---|---|---|
| 2 models | 59.87 | 5.60 | 6.70 | 6.99 | 1.59 | 6.94 | 6.45 | 1.52 | 6.96 |
| 3 models | 59.85 | 5.44 | 7.15 | 7.05 | 0.78 | 7.38 | 6.78 | 0.78 | 7.21 |
| 5 models | 59.89 | 5.07 | 6.68 | 6.83 | 0.30 | 6.95 | 6.46 | 0.35 | 7.21 |
| 7 models | 59.90 | 4.80 | 6.77 | 6.80 | 0.24 | 6.73 | 6.26 | 0.21 | 6.74 |
| 10 models | 59.90 | 4.81 | 6.86 | 6.92 | 0.13 | 6.92 | 6.22 | 0.11 | 6.95 |

## D.2  2 Models, 3 Models, 5 Models and 7 Models Accuracy V.S. Different Magnification Factors

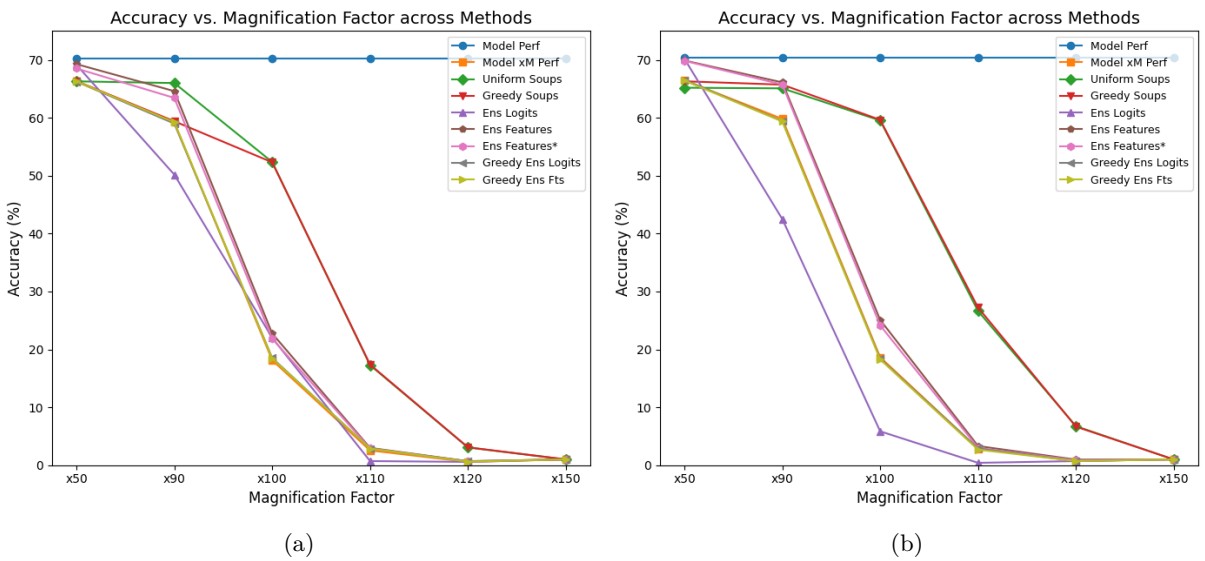

(a)                                                                (b)

Figure 23: (a) Accuracy vs. Magnification Factor across merging/ensembling methods with 2 models on CIFAR-100 dataset. (b) Accuracy vs. Magnification Factor across merging/ensembling methods with 3 models on CIFAR-100 dataset.

The 2 models, 3 models, 5 models and 7 models accuracy vs. different magnification factors on CIFAR-100 are shown via fig. 23a, fig. 23b, fig. 24a and fig. 24b, respectively.

Similar to CIFAR-100, the accuracy versus magnification factor on CIFAR-10 using 2, 3, 5, 7, and 10 models is shown in figs. 25a, 25b, 26a, 26b and 27a, respectively. Additionally, fig. 27b presents the accuracy gaps of uniform soup and logit ensembling under varying magnification factors across different model counts. Similar trends are observed as discussed earlier; however, on CIFAR-10, the performance gap peaks at a magnification factor of ×110, in contrast to ×100 for CIFAR-100.

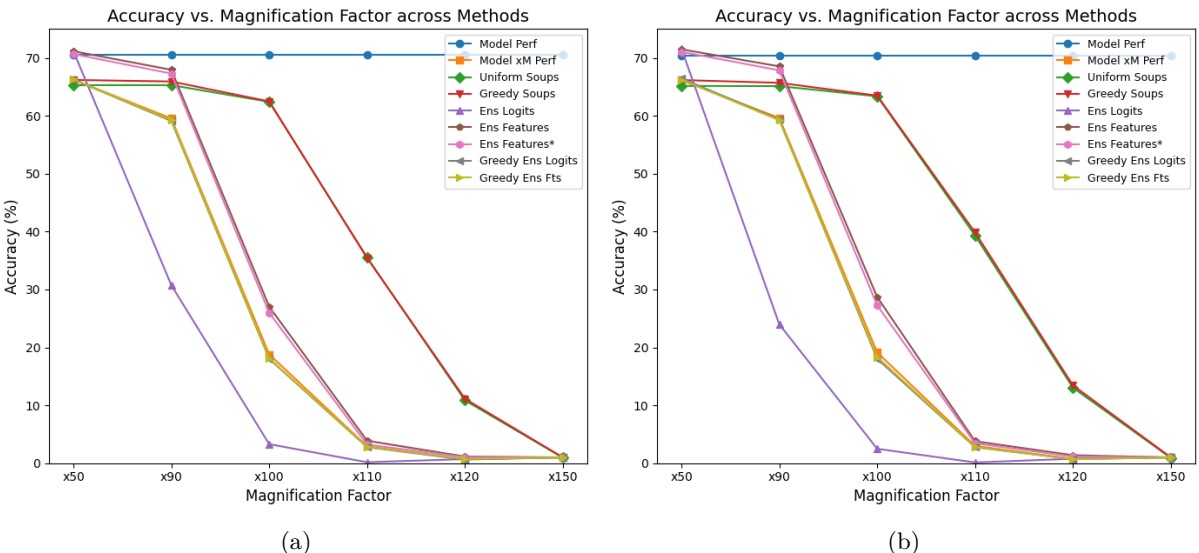

Figure 24: (a) Accuracy vs. Magnification Factor across merging/ensembling methods with 5 models on CIFAR-100 dataset. (b) Accuracy vs. Magnification Factor across merging/ensembling methods with 7 models on CIFAR-100 dataset.

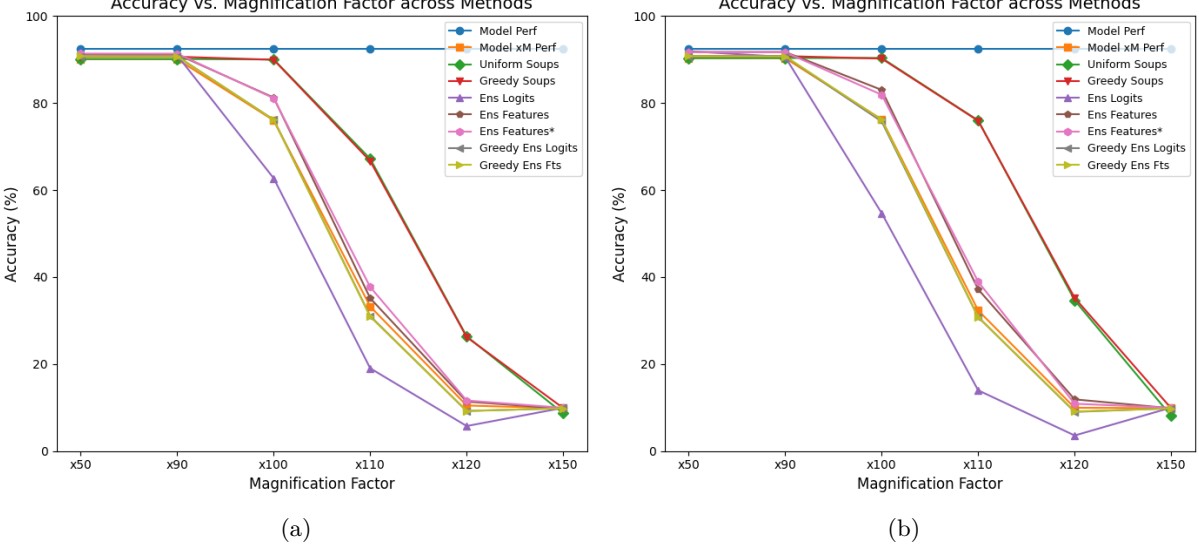

Figure 25: (a) Accuracy vs. Magnification Factor across merging/ensembling methods with 2 models on CIFAR-10 dataset. (b) Accuracy vs. Magnification Factor across merging/ensembling methods with 3 models on CIFAR-10 dataset.

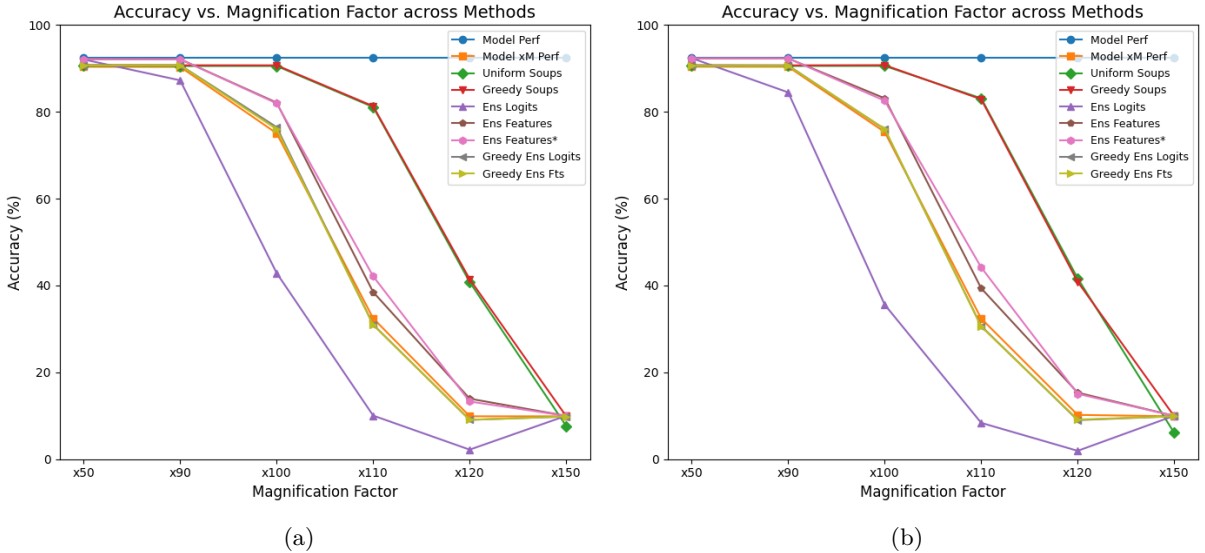

(a)                                                    (b)

Figure 26: (a) Accuracy vs. Magnification Factor across merging/ensembling methods with 5 models on CIFAR-10 dataset. (b) Accuracy vs. Magnification Factor across merging/ensembling methods with 7 models on CIFAR-10 dataset.

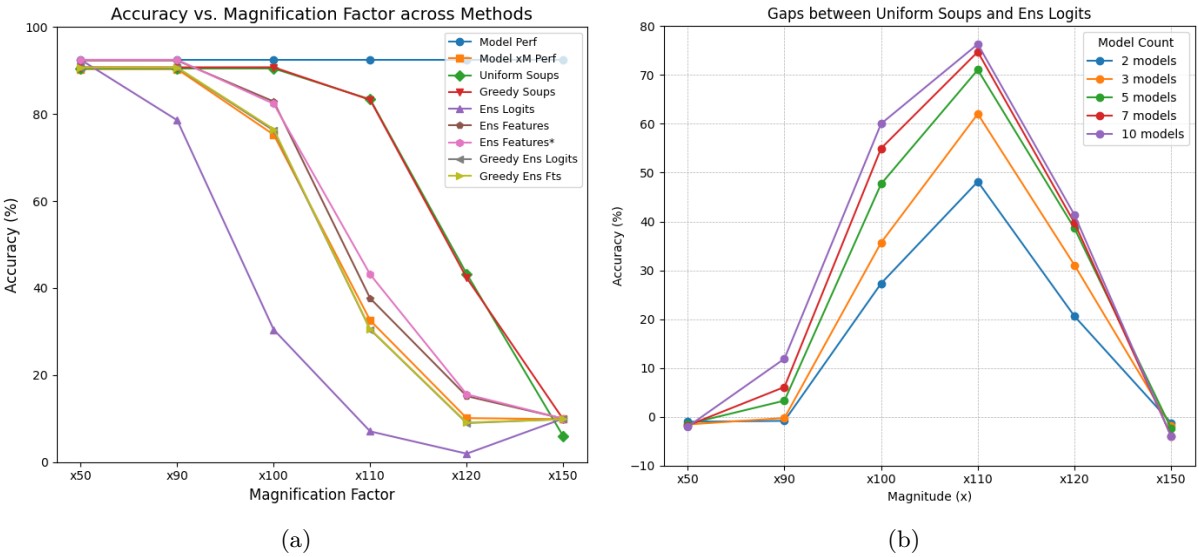

(a)                                                    (b)

Figure 27: (a) Accuracy vs. Magnification Factor across merging/ensembling methods with 10 models on CIFAR-10 dataset. (b) The gap between Uniform soup and Logits Ensemble Accuracy against Magnification Factor across different model numbers on CIFAR-10 dataset.

