# OpenReview forum: "Towards A More Transparent Understanding of Weight-Averaged Model Merging: A Qualitative and Quantitative Study"
_TMLR — Rejected by TMLR_

### Review · Reviewer_BuUd · 2025-11-20

**Summary Of Contributions:**

Summary:

This paper provides an investigation on weight-averaged model merging. The authors analyze why weight averaging works through and interesting template-matching perspectives, compare weight-space and feature-space averaging across a broad range of architectures, and study the impact of weight magnitude on robustness and generalization. Experiments are conducted on numerous image datasets and architectures.

Strengths:
- The paper is clearly written and well organized.
- The perspective on model averaging, particularly the template matching perspective, is interesting and offers a novel way to  study model merging.
- The experimental evaluation is extensive, covering a wide range of architectures (CNNs and ViTs) and multiple datasets.
- The comparative study between weight-space and feature-space averaging as well all the study on weight magnitudes/variances provides an interesting and underexplored explored point of view.

Weaknesses:
- Most of the analysis for deep models is put into the supplementary material. While this is understandable due to avoid making the paper too heave, including more discussion in the main paper would help ensure that readers do not have the wrong impression that the findings are limited to linear models.
- A dedicated Discussion section can improve the paper by discussing how these experimental observations could guide the design of future model-averaging techniques.
- The limitations of the study (e.g., sensitivity of ViTs to merging, constraints of the experimental setup) could be discussed more openly.

Overall, this is a valuable contribution.

Minor Comments:
- “However, from the standpoint of AI safety and alignment…”: please elaborate on the relevance to AI safety. If the connection is week, consider removing the term “AI safety” to avoid misunderstandings.
- Figure 4: Please explicitly label the axes as “Merged Model Accuracy” and “Ensembled Logits Accuracy” for clarity.
- Bibliography: Please, revise the references to ensure that published versions of papers are cited whenever available. For example, “Model merging by uncertainty-based gradient matching” should be cited as its ICLR 2024 version rather than the arXiv preprint. Many other entries appear to require similar corrections.

**Audience:**

Yes

**Audience Explanation:**

Model averaging is a timely topic and many people can find this paper interesting to guide future research in this field.

**Broader Impact Concerns:**

No Ethical Concerns

**Claims And Evidence:**

Yes

**Claims Explanation:**

There are many experiments on different computer vision datasets and models that supports the claims.

**Requested Changes:**

See the Weaknesses and Minor Comments

---

> ### Author Response · Authors · 2026-01-03
>
> **Q1:** Including more discussion in the main paper would help ensure that readers do not have the wrong impression that the findings are limited to linear models.
>
> **A1:** Thank you for the helpful suggestion. We agree that adding more discussion in the main paper can help avoid the incorrect impression that our findings are limited to linear models. Accordingly, we have expanded the discussion in Section 3 of the revised manuscript to explicitly clarify that our results also apply to non-linear models. We additionally provide further explanation to highlight why the presented analysis generalizes beyond the linear setting.
>
>
> **Q2:** A dedicated Discussion section can improve the paper by discussing how these experimental observations could guide the design of future model-averaging techniques.
>
> **A2:** We appreciate this suggestion. In the revised manuscript, we have added a dedicated Discussion section that highlights how our experimental findings can inform the design of future model-averaging methods. Specifically:
>
> (a) We discuss how the identified structured weight patterns in both CNNs and ViTs suggest that future merging techniques may benefit from structure-aware or representation-aligned averaging strategies.
>
> (b) We elaborate on how the observed differences between weight-space merging and feature-space ensembling indicate that hybrid or task-adaptive merging schemes may be more effective than uniform averaging.
>
> (c) We describe how our analysis of parameter scaling and prediction stability implies that future methods could explicitly incorporate scale-normalization or regularization mechanisms to enhance robustness.
>
> We also mention Broader Impact and Limitation in the discussion. We believe that this new Discussion section strengthens the paper.
>
>
> **Q3:** The limitations of the study (e.g., sensitivity of ViTs to merging, constraints of the experimental setup) could be discussed more openly.
>
> **A3:** Thank you for this valuable comment. We agree that a clearer discussion of the study’s limitations would improve the transparency of the work. In the revised version, we have added a dedicated paragraph in the Discussion section to more openly acknowledge the limitations.
>
>
> **Q4:** Minor Comments.
>
> **A4:** We appreciate the reviewer’s constructive minor comments. We have carefully addressed all of the minor issues raised, and the corresponding revisions have been incorporated into the revised manuscript.

---

### Review · Reviewer_rv59 · 2025-11-29

**Summary Of Contributions:**

This paper investigates weight-averaged model merging from an interpretability standpoint, asking why and when simple parameter averaging produces useful models. The authors organize their study around three perspectives: (1) viewing learned weights as class-specific templates, (2) comparing weight-space averaging against feature-space averaging (i.e., ensembling), and (3) analyzing how weight magnitude and variance affect merged model predictions. Experiments span multiple architectures and datasets.

**Strengths**
The paper tackles a real gap in our understanding. Weight averaging has become a standard method but we know exactly how it works The safety/interpretability framing is reasonable given the increasing deployment of merged models.

The template visualization for linear classifiers (Figures 1-2) is nice. Showing that classifier weight vectors resemble class-averaged images, and that averaging weights blends the representations, gives a concrete mental model for what's happening. The Mixup analogy is apt here.
The experiments are comprehensive. Testing across several datasets with varying numbers of merged models (2-10) provides a fairly complete picture of the phenomena. I appreciate that the authors include both standard vision benchmarks and medical imaging datasets.

The finding that ViTs are much more fragile under weight averaging than CNNs (Figure 3, Table 9) is important. The performance of uniform soups drops as more ViT models are merged, while greedy selection provides no benefit beyond the best single model. This matches observations from practitioners and deserves more more analysis both theoretically and impirically.
The magnitude/variance analysis (Section 5) makes sense: averaging reduces the maximum weight magnitude and variance, which bounds output variance and provides implicit regularization. The scatter plots in Figure 4 showing that merging maintains reasonable accuracy even at extreme scaling factors (where ensembling collapses) are interesting.

**Weaknesses**
The main issue is the gap between what's claimed and what's demonstrated. The paper promises to explain "why model merging works," but mostly provides intuitions and visualizations rather than mechanisms. The template story is compelling for a single linear layer, but the extension to deep networks is essentially "the same thing happens at each layer," which isn't really an explanation.

Remark 1 states that poor merging may arise from "conflicts between individually learned patterns." This is exactly the right question to ask, but the paper never actually measures or characterizes these conflicts. Can we predict, before merging, whether two models will combine well? What features of the weight matrices indicate compatibility? This is where I was hoping for a contribution, but it doesn't come.

The relationship to mode connectivity and loss landscape flatness is absent. Weight averaging works when models lie in the same space; the template matching perspective doesn't really address when or why this happens. Git Re-Basin and related work get a mention in the related work section but aren't evaluated with experimentally.

The weight-versus-feature comparison (Section 4) establishes things that are already known. For linear models they're equivalent; for nonlinear models they differ due to the placement of activations. That ensembling generally beats merging isn't surprising, the interesting question is characterizing the gap, which doesn't get much attention.

The paper doesn't compare against TIES-Merging, DARE, Fisher Merging, or other recent methods that explicitly handle weight conflicts. Given that the paper is about understanding merging, engaging with methods designed to address merging failures seems important.

**Minor Points**
-The paper uses "interpretability" when it often means "visualization", these aren't the same thing.
-Property 1 and Theorem 1 are from Wang et al. (2024); this should be clearer in the main text rather than just the appendix.
-Error bars would help, especially in Table 2 where some differences are small.
-The notation in Section 5 switches between W bar and W.

Questions for Authors

1.Can you measure template similarity (e.g., cosine similarity between corresponding weight rows) across random seeds and correlate this with merge success?
2.For the ViT failure mode: is this about attention head specialization, LayerNorm statistics, or something else? Even a preliminary investigation would strengthen the paper.
3.Does the regularization effect of merging actually help OOD generalization, or only in-distribution accuracy?

This is a solid empirical study with useful observations, particularly regarding ViT fragility and the robustness properties of merged models. The visualizations are helpful for building intuition. However, the paper oversells its contributions, it provides descriptions and visualizations rather than mechanistic explanations. The template matching lens is intuitive but doesn't yet explain when merging will succeed or fail.
I'd suggest the authors reframe the contribution as "an empirical investigation with interpretable visualizations" rather than "explaining why merging works." With that adjustment and some engagement with the mode connectivity perspective, this would be a nice contribution.

**Audience:**

Yes

**Audience Explanation:**

The paper addresses a topic of clear practical relevance. Weight-averaged model merging has become a widely used technique across multiple domains.
Despite this widespread adoption, there is relatively little systematic empirical analysis of when and why these methods work. Practitioners regularly make decisions about whether to merge models without strong guidance on what to expect.
Several of the paper's findings would be directly useful to this audience. The observation that Vision Transformers are substantially more fragile under weight averaging than CNNs (with uniform soup performance degrading sharply as more models are merged) is practically important and not widely documented. Researchers working with ViTs should be aware of this limitation.
The systematic comparison between weight averaging and feature averaging across architectures and datasets provides a useful reference. While the general finding that ensembling outperforms merging is expected, the quantification of the gap across different settings and the identification of cases where merging performs particularly poorly (e.g., PathMNIST, CelebA with uniform soups) gives practitioners concrete guidance.
The template visualization approach, while not a complete mechanistic explanation, offers an accessible mental model for reasoning about what weight averaging does. This kind of intuition-building has value for researchers designing new merging strategies or debugging failures.

The paper also raises important questions that could stimulate follow-up work: Why do ViTs fail at merging? Can we predict merge compatibility before combining models? How does the template perspective connect to mode connectivity? These open questions are themselves valuable contributions to the discourse.
Overall, while the paper's explanatory claims outpace its evidence, the empirical findings and systematic comparisons would be of interest to researchers and practitioners working on model merging, ensemble methods, and transfer learning.

**Broader Impact Concerns:**

The broader impact discussion could be strengthened. Since weight-averaged model merging is increasingly used in practical ML systems (including safety-critical ones), understanding its failure modes is essential. The paper touches on interpretability but does not fully address the risks associated with merging incompatible models, particularly in settings where merging might silently degrade performance on minority subpopulations or create unpredictable behavior in downstream tasks.

Additionally, the work highlights that Vision Transformers merge poorly, but the potential safety implications of this fragility are not discussed. If practitioners assume that merging provides “free” ensembling-like benefits, these failure modes could lead to overconfidence or unintended deployment risks.

Overall, the paper would benefit from explicitly discussing scenarios where weight merging may introduce harm or instability, and outlining best-practice guidelines.

**Claims And Evidence:**

No

**Claims Explanation:**

The paper's central claim, to explain "why and how weight-averaged model merging works", is not fully supported by the evidence provided. While the empirical observations are thorough and the visualizations are informative, there is a persistent gap between the descriptive findings and the explanatory claims.

The template matching perspective is intuitive for linear classifiers, where visualizing weight vectors as class templates is straightforward. However, the extension to deep nonlinear networks relies on general statements like "deeper layers detect different features" without rigorous validation. The paper does not demonstrate that the template interpretation actually holds in intermediate layers of deep networks, nor does it show how template alignment (or misalignment) predicts merging success.
Remark 1 asserts that suboptimal merging results from "conflicts between individually learned patterns," but this hypothesis is never operationalized or tested. The paper provides no metric for measuring such conflicts, no experiments correlating conflict measures with merge outcomes, and no way to predict a priori whether two models will merge well. This is the central mechanistic question, and it remains unanswered.
The weight-versus-feature averaging analysis (Section 4) correctly notes that the two are equivalent for linear models and differ for nonlinear ones, but this is well-established. The finding that ensembling generally outperforms merging is expected and does not constitute new insight into the mechanisms at play.
The magnitude scaling experiments (Section 5) demonstrate that weight averaging is more robust than ensembling under extreme parameter scaling. The connection to variance reduction is mathematically sound, but the practical relevance of scaling weights, and the claim that merging "acts like regularization" is not tested against actual generalization metrics or out-of-distribution performance.

Finally, the paper does not engage with the mode connectivity and loss landscape literature in any experimental way, despite this being the most direct theoretical framework for understanding when weight averaging succeeds. The template perspective is offered as an alternative lens but is not connected to these established concepts.

In summary, the paper provides useful empirical characterizations and visualizations, but the evidence does not rise to the level of explaining the underlying mechanisms of model merging. The claims would be better supported if framed as "empirical observations and interpretable visualizations" rather than mechanistic explanations.

**Requested Changes:**

1. Clarify the scope of the paper’s claims.
The current framing suggests that the paper explains why weight merging works in general. The explanations are largely intuitive or descriptive, so the paper should temper its language and present itself as an empirical/interpretability-driven investigation instead of a mechanistic theory.

2. Improve the discussion of related work: especially mode connectivity, permutation alignment, and modern merging methods.
The paper briefly cites these works but does not deeply connect the empirical findings to this literature. A stronger positioning is needed to situate the template-matching perspective relative to known explanations for merging success/failure.

3. Add at least some analysis of “template compatibility” or “pattern conflicts.”
Since this is one of the paper’s central explanations for merging behavior, readers need a more concrete way to understand or measure such conflicts. Even simple metrics (e.g., weight similarity, CKA alignment, correlation across seeds) would make the argument more grounded.

4. Clarify the role of external theoretical results.
Property 1 and Theorem 1 come from prior work (Wang et al. 2024). The paper should explicitly state that these results are not contributions of this work, and explain how they support (rather than constitute) the paper’s analysis.

Minor:
Deepen the analysis of Vision Transformer fragility.
This is one of the most interesting empirical findings, and further discussion (e.g., attention head specialization, layer norm sensitivity, permutation symmetry issues) would significantly enhance the paper.

Provide either error bars or variability estimates across seeds for the main tables and plots, given the stochastic nature of training.
Discuss practical implications of weight-scaling experiments.

Since the extreme scaling regimes may not arise naturally in training, it would help to situate these findings more clearly in practical contexts (e.g., robustness to rescaling in federated or multitask settings).

Polish notation and clarity.
Ensure consistent notation in Section 5 (W bar and W)  and make the distinction between “interpretability” and “visualization” clearer throughout.

---

> ### Author Response · Authors · 2026-01-03
> **Responses to Reviewer rv59 (Part 1)**
>
> **Q1:** The paper mostly provides intuitions and visualizations rather than mechanisms.
>
> **A1:** We use visualizations to aid intuition, but the core contributions of the paper are explanatory novel point of views and qualitative/quantitative analyses that provide interpretability and a more transparent understanding of weight-averged model merging (as the title). We do not equate interpretability with visualization; instead, visualization is one auxiliary tool supporting our qualitative and quantitative arguments.
>
> In response to the reviewer’s comments, we have substantially revised the manuscript in multiple sections to better reflect that the paper is centered on qualitative and quantitative analysis. Please refer to the update paper for more details.
>
>
> **Q2:** Remark 1 states that poor merging may arise from "conflicts between individually learned patterns." This is exactly the right question to ask, but the paper never actually measures or characterizes these conflicts.
>
> **A2:** Thank you for raising this point. Our intention in Remark 1 was not to claim that we fully measure or enumerate all forms of “conflicts between individually learned patterns,” but rather to highlight, for the very first time that presents formally in a paper to the our best knowledge, that model merging results can be meaningfully understood through a template-matching interpretability perspective.
>
> Thus, it seems there may have been a misunderstanding: the remark is meant to motivate new interpretability, not to promise an exhaustive taxonomy or quantitative characterization of all possible conflicts. Our analyses of the 1st part focus on establishing (1) that structured, interpretable weight templates exist and (2) that such structure provides a principled way to understand both successful and unsuccessful merging behaviors. Importantly, our findings show that both CNNs and ViTs exhibit structured and semantically meaningful weight patterns, suggesting that averaging procedures may benefit from explicitly leveraging these structures. A full systematic analysis of every type of conflict is beyond the scope of this work, but they are very interesting and they can be follow-up studies in the future work.
>
>
>
> **Q3:** The weight-versus-feature comparison (Section 4) establishes things that are already known.
>
> **A3:** Thank you for the comment. While prior work has compared weight averaging with output-space ensembling in broad terms, our study provides several novel, previously unreported findings under a unified and carefully controlled extensive qualitative and quantitative experimental framework. These reveal new findings in how different merging strategies behave across architectures, depths, and merging cardinalities. Specifically:
>
> (1) Under identical experimental conditions, Vision Transformers (ViTs) show substantially larger performance gaps between uniform weight averaging and logits ensembling. Moreover, these gaps widen as more models are included, revealing architecture-dependent sensitivity.
>
> (2) As the number of merged models increases, the performance gap between logits ensembling and uniform soups grows as well. This scaling behavior is not reported in prior work and provides actionable guidance for practitioners when merging many models.
>
> (3) Accuracy gaps vary systematically across model families and tend to shrink as depth increases, likely due to representational saturation in deeper networks. We also observe that batch normalization disrupts this pattern, another insight not previously established.
>
> (4) Greedy soups mitigate these gaps, narrowing the discrepancy between merging and ensembling. This demonstrates that strategic selection is crucial for stabilizing merging outcomes—again, a nuance that is not part of existing canonical knowledge.
>
> The abovementioned findings are novel, and experimentally grounded. Our statement in Section 4 is not simply to restate well-known facts, but to provide a fine-grained qualitative and quantitative analysis that exposes new empirical regularities and architectural differences that were not previously recognized.

---

> ### Author Response · Authors · 2026-01-03
> **Responses to Reviewer rv59 (Part 2)**
>
> **Q4:** The paper doesn't compare against TIES-Merging, DARE, Fisher Merging, or other recent methods that explicitly handle weight conflicts. Given that the paper is about understanding merging, engaging with methods designed to address merging failures seems important.
>
> **A4:** The intention of the paper is not to put thorough evaluations/comparisons against recent methods that handle weight conflicts, but offering novel point of views to look at weight-averaged model-merging (which is also indicated in the paper title). In order to emphasize it, we actually have put a dedicated paragraph in the original paper after the contribution section. “The intention of the paper: Instead of providing a thorough evaluation of existing works, as a survey paper, this paper focuses on an empirical investigation of why and how weight-averaged model merging can be effective, through the three interpretability-driven perspectives outlined above. Grounded in our empirical findings, we further provide practical proposals and insights that aim to guide future work and support a more transparent understanding of model merging within the broader AI alignment context.” Besides, we also rewrite and strengthen the part to discuss the related work in the revised version of the paper.
>
>
> **Q5:** The paper uses "interpretability" when it often means "visualization".
>
> **A5:** Thank you for the comment. We would like to clarify that our analysis is not limited to visual inspection. Visualization is only one tool we employ (we have extensive quantitative experiments and findings as well), but it is not the substance of our interpretability claims. Our goal is to provide three novel and complementary perspectives that explain why weight-averaged model merging works. This perspective allows us to interpret the merged model’s behavior in terms of representational structure.
>
> In terms of these 3 view points, specifically,
>
> (a) Template-matching perspective (novel interpretability viewpoint).
> We examine how individual models implicitly learn class-specific templates or prototypes encoded in their weights, and how these templates interact under weight averaging. This perspective allows us to interpret the merged model’s behavior in terms of representational structure, not merely its visual appearance. To our knowledge, we are the first to interpret weight-averaged merging explicitly through this template-matching lens.
>
> (b) Weight-space vs. feature-space averaging analysis.
> This part of the paper provides a functional interpretability comparison: via extensive qualitative experiments, we analyze how merging in different spaces affects representations and downstream predictions. This is an interpretability question about mechanisms, not a visualization exercise.
>
> (c) Parameter-scaling and stability analysis.
> Also, via many qualitative experiments and analyses, we study how scaling affects prediction stability and weight averaging can behave like a regularizer. This offers insight into the model’s optimization and robustness properties. Also, it is a mechanistic interpretability angle rather than visualization.
>
>
> **Q6:** Can you measure template similarity (e.g., cosine similarity between corresponding weight rows) across random seeds and correlate this with merge success?
>
> **A6:** Thank you for this suggestion. As requested, we now measure template similarity across random seeds and analyze its relationship with merge success. We have added a detailed investigation in Section 6.2.2, “Model Similarity v.s. Accuracy of Merged Models.”
>
> Specifically, we examine two complementary levels of similarity:
> (1) Model-wise similarity, computed as the average cosine similarity between corresponding classifiers across seeds, and
> (2) Class-wise similarity, where cosine similarity is measured for each individual class template and correlated with the corresponding per-class accuracy of the merged model.
>
> Across both analyses, we observe a strong positive correlation between template similarity and merge performance, demonstrating that better-aligned templates across random seeds lead to more successful model merging. These results provide quantitative support for the role of template alignment in explaining merge behavior.

---

> ### Author Response · Authors · 2026-01-03
> **Responses to Reviewer rv59 (Part 3)**
>
> **Q7:** A preliminary investigation of the reason in ViT merging failure would strengthen the paper.
>
> **A7:** We thank the reviewer for the comments. We’ve stated the reason in the Section 6.3.1 of the original paper: “Notably, Vision Transformer (ViT) architectures exhibit significantly larger performance gaps between uniform weight averaging and logits ensembling, and these gaps widen as more models are added. This may stem from fundamental architectural differences between convolutional neural networks (CNNs) and ViTs. CNNs are inherently suited for learning hierarchical local patterns, making them more stable under weight-averaged merging. In contrast, ViTs rely on global self-attention mechanisms that capture token-level dependencies, rendering their learned representations more sensitive to perturbations in weights. Similar performance degradation for ViTs is observed on other datasets as well, such as the DeiT model on TinyImageNet (see supplementary materials for details).”
>
>
> **Q8:** Property 1 and Theorem 1 are from Wang et al. (2024); this should be clearer in the main text rather than just the appendix.
>
> **A8:** Thank you for the comments. We have already stated in the original main text of Section 5.2 that “We have the following property 1 and theorem 1~\cite{wang2024corelocker} to build the connections between model weights' magnitudes/variances and outputs' magnitudes/variances”. I hope these texts answered your question.
>
>
> **Q9:** The notation in Section 5 switches between W bar and W.
>
> **A9:** Thank you for the comments. In our paper, $\bar{W}$ and $W$ denote different quantities: $\bar{W}$ represents the weight tensors after merging, whereas $W$ refers to the original weights.
>
>
> **Q10:** The broader impact discussion could be strengthened.
>
> **A10:** We thank the reviewer for the suggestion. We’ve added one paragraph to talk about the broader impact in the discussion section. Please refer to the updated version of the paper.

---

### Review · Reviewer_dZRX · 2025-12-21

**Summary Of Contributions:**

This manuscript studies why weight averaging works, presenting empirical analysis in interpretability perspectives. The authors present analyses with three directions, such as template matching structure analysis, comparing weight averaging with feature averaging, and scaling pretrained weights for implicit regularization.

**Audience:**

Yes

**Audience Explanation:**

Weight averaging is widely used in the machine learning field; several researchers would find certain values in this study.

**Claims And Evidence:**

No

**Claims Explanation:**

I think current version is not sufficient. Please see the Requested Changes below.

**Requested Changes:**

- The target datasets are mainly the CIFAR dataset, which is small scale. In fact, CIFAR images are 32x32 in size, which requires different architectural configurations on ResNet and ViT. For example, ResNet for CIFAR datasets commonly uses three stages, not the four stages, to be compatible with the 32x32 size and to avoid extensive downsampling. Similarly, I wonder whether applying and studying ViT-B/16 for the CIFAR dataset is appropriate. Applying extensive upsampling to CIFAR to be compatible with 224x224 looks inappropriate.
- Although the authors present several qualitative explanations for merging, the presented analysis lacks quantitative analysis to investigate and diagnose why merging succeeds and when merging fails. It would be interesting to establish certain indicators to quantify them.
- Several recent merging methods, such as TIES and Fisher, which are mentioned in related works, are not evaluated in experiments. The targeted merging methods are limited to specific methods.
- The failure in ViT merging is interesting, but the core cause of the failure is not presented.
- Property 1 and Lemma 1 have different definitions for $K_s$, especially for the row operation.
- The authors wrote “bounded by the max entry value of weight matrices.” but the constant $K_s$ does not indicate entrywise maximum.
- On page 16 in the appendix, the authors wrote, “By the Lipschitz property of the activation function, we have …” and use $\phi(u) \leq L u$, but the correct Lipschitz property would be $|\phi(u)-\phi(v)| \leq L |u-v|$. The current derivation seems incorrect.
- On page 15, $\phi(0)=0$ is used without explanation, but it is not convincing.
- Eq. 12 should be revised: the index $m$ should be used within the product, but the term $(L^2 N)^m$ is outside of the product. The index $l$ is not used in the product.
- On page 15, the authors compute a probability of $||Av||_2 \leq s A ||v||_2$, but I think that this is always true to yield 1, because $||Av||_2 \leq ||A||_2 ||v||_2 = s_1 A ||v||_2$. Please check this point.
- I have checked the source code, but several parts are incorrect.
    - In cifar10_merge.py, the authors implement something like candidate_soup = (best_soup + state_dict) / 2, which, however, does not correctly implement uniform average.
    - The authors apply magnitude scaling using state_dict.items(). However, state_dict.items() contains others, such as running_mean in batch normalization; correct implementation should use model.named_parameters() or something.
    - For test_ens_fts(), please check whether the code applies average on the classifier.
- Writing should be improved.
    - “For Eq. 14 in the paper” → “For Eq. 10 in the paper”
    - “For example, Wortsman et al. Wortsman et al. (2022) proposed” → “For example, Wortsman et al. (2022) proposed”
    - “How Weight Magnitudes and Variance Effects Model Outputs” → “How Weight Magnitudes and Variance Affects Model Outputs”
    - “Lipschiz” → “Lipschitz”

---

> ### Author Response · Authors · 2026-01-03
> **Responses to Reviewer dZRX (Part 1)**
>
> **Q1:** The target datasets are mainly the CIFAR dataset, which is small scale.
>
> **A1:** We thank the reviewer for the comment. We emphasize that the goal of our work is not architectural optimization for CIFAR, but to study the behavior of weight-averaged model merging under commonly used architectures as long as they are kept in a fair comparison. For ResNet, while CIFAR-specific variants with fewer stages are sometimes used, our conclusions do not depend on such architectural details. We intentionally adopt standard ResNet configurations to keep the setup consistent with prior model-merging and weight-averaging studies, and to isolate the effects of merging rather than architectural tuning.
>
> For ViT, although CIFAR images are upsampled to 224×224 to match ViT-B/16, this practice is standard in the literature when analyzing ViT behavior on CIFAR and small datasets. Our focus is on relative trends and stability properties, not absolute performance. Using ViT-B/16 allows us to test whether our observations generalize beyond CNNs to transformer architectures with different inductive biases.
>
> Moreover, CIFAR is not the only benchmark in our study (please refer to 6.1 Experimental Settings). We conduct experiments on multiple datasets of varying scales (including much larger settings, e.g. CelebA 178×218, Tiny ImageNet 64 × 64, Chest X-Ray 14 Dataset 1024×1024), and observe consistent trends across datasets and architectures. This supports that our conclusions are not only on CIFAR resolution or specific architectural choices.
>
>
> **Q2:** Although the authors present several qualitative explanations for merging, the presented analysis lacks quantitative analysis to investigate and diagnose why merging succeeds and when merging fails. It would be interesting to establish certain indicators to quantify them.
>
> **A2:** We thank the reviewer for the comments. In addition to conceptual explanations, the paper contains quantitative analyses: For example, Section 6.3 shows Weights vs. Features Averaging over CNNs or ViTs on different datasets, and we based on the results qualitatively analyze when the merging can or cannot outperform ensembling. Section 6.4 provides quantitative studies on how prediction stability and performance vary under magnitude scaling, directly diagnosing conditions under which weight-averaged merging remains stable or degrades. These sections go beyond qualitative intuition by introducing measurable quantities (e.g., performance trends under controlled scaling and perturbation) that characterize the success and breakdown of merging. We will further emphasize and clarify these quantitative diagnostics in the revised version to make their role more explicit. Establishing indicators to quantify them is an interesting topic, and we view it as an important future work.
>
>
> **Q3:** Several recent merging methods, such as TIES and Fisher, which are mentioned in related works, are not evaluated in experiments. The targeted merging methods are limited to specific methods.
>
> **A3:** The intention of the paper is not to put thorough evaluations/comparisons against recent methods that handle weight conflicts, but offering novel point of views to look at weight-averaged model-merging (which is also indicated in the paper title). In order to emphasize it, we actually have put a dedicated paragraph in the original paper after the contribution section. “The intention of the paper: Instead of providing a thorough evaluation of existing works, as a survey paper, this paper focuses on an empirical investigation of why and how weight-averaged model merging can be effective, through the three interpretability-driven perspectives outlined above. Grounded in our empirical findings, we further provide practical proposals and insights that aim to guide future work and support a more transparent understanding of model merging within the broader AI alignment context.” Besides, we also rewrite and strengthen the part to discuss the related work in the revised version of the paper.

---

> ### Author Response · Authors · 2026-01-03
> **Responses to Reviewer dZRX (Part 2)**
>
> **Q4:** The failure in ViT merging is interesting, but the core cause of the failure is not presented.
>
> **A4:** We thank the reviewer for the comments. We’ve stated the reason in the Section 6.3.1 of the original paper: “Notably, Vision Transformer (ViT) architectures exhibit significantly larger performance gaps between uniform weight averaging and logits ensembling, and these gaps widen as more models are added. This may stem from fundamental architectural differences between convolutional neural networks (CNNs) and ViTs. CNNs are inherently suited for learning hierarchical local patterns, making them more stable under weight-averaged merging. In contrast, ViTs rely on global self-attention mechanisms that capture token-level dependencies, rendering their learned representations more sensitive to perturbations in weights. Similar performance degradation for ViTs is observed on other datasets as well, such as the DeiT model on TinyImageNet (see supplementary materials for details).”
>
>
> **Q5:** Property 1 and Lemma 1 have different definitions for Ks.
>
> **A5:** Thank you for pointing this out. We agree that there is an inconsistency in the current draft. We followed Lemma 1, Property 1 and Theorem 1 from work (Wang et al. 2024). In the revision, the definitions have been improved consistently. We also correct the description. The statement has been changed to say that the output norm growth is controlled by the Lipschitz constant of the activation function and an upper bound on the spectral norm of the weight matrices (obtained from Lemma 1), instead of the max entry value.
>
>
> **Q6:** On page 16 in the appendix, the authors wrote, “By the Lipschitz property of the activation function, we have …” The current derivation seems incorrect.
>
> **A6:** Thanks for flagging this. In the first layer, $u_i^{(1)} = W^{*,(1)}x + b_i^{(1)}$. To make the Lipschitz step explicit (and independent of the sign of $u_i^{(1)}$), we will add the standard absolute-value sign, and its form anchored at $0$ (since $\phi(0)=0$):$|\phi(u_i^{(1)})-\phi(0)| \le L,|u_i^{(1)}-0|.$ This is the version we rely on, because the rest of the argument works with second-order/variance bounds where only the magnitude matters (the sign is absorbed). Adding this step is purely clarifying and does not change any downstream inequalities or the final conclusion.
>
>
> **Q7:** On page 15, φ(0) = 0 is used without explanation, and it is not convincing.
>
> **A7:** Thank you for the question. In the current appendix, the step ||φ(z)|| = ||φ(z) − φ(0)|| implicitly assumes φ(0) = 0, which should be explicitly mentioned. In the revised version, we will explicitly state that φ(0) = 0 in Property 1. This holds for commonly used activation functions in our experiments, such as ReLU, GELU, and tanh. Under this assumption, the Lipschitz-based bound follows directly.
>
>
> **Q8:** Equation 12 should be revised. The index m should be used within the product, but the term (L^2 N)^m is outside of the product. The index l is not used in the product.
>
> **A8:** Thank you for pointing this out. We will revise by absorbing $\left(L^2 N\right)^M$ into the product, e.g., $\|\boldsymbol{x}\|_2^2 \prod_{k=1}^M\left[L^2 N\left(\sigma_w^{(k)}\right)^2\right]$. The bound itself remains unchanged.
>
> $l$ is just a dummy index for the inner product over layers $m + 1$ to $M$; it appears in $(\sigma_w^{(l)})$.
>
>
> **Q9:** On page 15, the authors compute a probability of ||A v||_2 ≤ s1(A) ||v||_2, but this inequality is always true and should yield probability 1.
>
> **A9:** In our proof, the probability comes from the random-matrix assumption on A (sub-Gaussian weights): we bound |A|_2 by a constant with high probability, and then the deterministic inequality implies |Av|_2 \le (\text{that bound})|v|_2. We will clarify this to avoid conflating the probabilistic operator-norm bound with the deterministic inequality.
>
>
> **Q10:** In cifar10_merge.py, the authors implement something like candidate_soup = (best_soup + state_dict) / 2, which, however, does not correctly implement uniform average.
>
> **A10:** Thank you for the question. We’ve also provided the greedy_soup_devideLen() function, not sure if this is the one you are looking for.

---

> ### Author Response · Authors · 2026-01-03
> **Responses to Reviewer dZRX (Part 3)**
>
> **Q11:** The authors apply magnitude scaling using state_dict.items(). However, state_dict.items() contains others, such as running_mean in batch normalization; correct implementation should use model.named_parameters() or something.
>
> **A11:** Thank you for raising this point. We would like to clarify that the use of state_dict.items() is intentional rather than an oversight. In our implementation, we intentionally apply magnitude scaling to all entries in the model state, including Batch Normalization statistics such as running_mean and running_var. Our goal is to scale the entire model state consistently, rather than restricting the operation to learnable parameters only. From our perspective, Batch Normalization statistics are an integral part of the model’s functional behavior, and excluding them would introduce an inconsistency between scaled parameters and unscaled normalization states.
>
> Importantly, the original Model Soups (which is the main focus of our analysis) paper does not state that Batch Normalization parameters or buffers should be excluded from merging or scaling. As a result, we consider merging BN-related states to be a valid and reasonable design choice under the model merging framework. Our approach follows the same principle of treating the model as a whole, without selectively omitting specific components unless explicitly justified.
>
> We will clarify this design choice in the revised version of the paper to avoid confusion, and explicitly state that our magnitude scaling is applied to the full state_dict, including Batch Normalization buffers.
>
>
> **Q12:** For test_ens_fts(), please check whether the code applies average on the classifier.
>
> **A12:** Thank you for the comments. Please refer to the “ft_ave = torch.mean(torch.stack(features_list), dim=0)” in the test_ens_fts() function.

---

### Review · Reviewer_zfuv · 2025-12-21

**Summary Of Contributions:**

## Summary:
This paper introduces an interpretability-driven analysis of the model merging techniques, which averages multiple pre-trained weights to improve the robustness of predictions. The authors argue that model weights encode structured, template-like representations, and show through visualizations and theory that averaging weights corresponds to a meaningful linear combination of these templates. Such a simple merging can reduce parameter magnitude and variance, leading to more stable and robust predictions. The paper provides a theoretical analysis of the relationship between parameter and feature merging and also shows its influence on parameter distribution.

## Strengths:
Model merging is an important technique for improving model performance. This work investigates why model merging is effective, and through visualizations on small-scale datasets together with quantitative experimental analyses, shows that model merging can achieve performance comparable to that of model ensembling.

## Weaknesses:

One major limitation of this work is that it does not explicitly demonstrate why model merging leads to strong empirical performance. For example, although the paper claims that model merging reduces parameter variance, the connection between this reduction and actual performance improvements is not shown to be strongly or directly correlated. In addition, the template-based analysis provides limited new insight, since convolutional kernels/linears inherently function as pattern-matching operators.

**Additional Comments:**

N/A

**Audience:**

Yes

**Audience Explanation:**

Model merging is an important and widely-used technique.

**Claims And Evidence:**

No

**Claims Explanation:**

The claims in this work are mainly supported by visualizations of learned weights and empirical results on relatively small datasets (e.g., CIFAR). However, it remains unclear how model merging concretely alters the model’s decision-making process. Moreover, to the best of my knowledge, model merging tends to perform poorly when the participating models exhibit a significant parameter gap, yet this important scenario is not investigated in the paper.

**Requested Changes:**

1. It would be very helpful if the authors could explore the effectiveness of model merging on larger-scale datasets and a broader range of tasks, rather than focusing primarily on parameter merging for 32×32 images, where the learned model parameters are likely to be relatively similar.
2. Additional experiments on model merging under substantial parameter discrepancies would strengthen the paper—for example, by training models on two significantly different tasks and then evaluating the outcomes of model merging.
3. More direct evidence is needed to demonstrate that reducing parameter variance indeed leads to improved model performance.

---

> ### Author Response · Authors · 2026-01-03
>
> **Q1:** It would be very helpful if the authors could explore the effectiveness of model merging on larger-scale datasets and a broader range of tasks, rather than focusing primarily on parameter merging for 32×32 images, where the learned model parameters are likely to be relatively similar.
>
> **A1:** We thank the reviewer for the comment. We emphasize that CIFAR is not the only benchmark in our study (please refer to 6.1 Experimental Settings). We conduct experiments on multiple datasets of varying scales (including much larger settings, e.g. CelebA 178×218, Tiny ImageNet 64 × 64, Chest X-Ray 14 Dataset 1024×1024), and observe consistent trends across datasets and architectures. This supports that our conclusions are not only on CIFAR resolution or specific architectural choices.
>
>
> **Q2:** Additional experiments on model merging under substantial parameter discrepancies would strengthen the paper—for example, by training models on two significantly different tasks and then evaluating the outcomes of model merging.
>
> **A2:** Thank you for the comments. We would like to highlight that experiments under substantial parameter discrepancies are already included in Table 3. In this setting, Model1 is a VGG19 trained and evaluated on CIFAR-100 (70.37% accuracy), while Model2 is trained on a different dataset/task but evaluated on CIFAR-100.
>
>
> **Q3:** More direct evidence is needed to demonstrate that reducing parameter variance indeed leads to improved model performance.
>
> **A3:** Thank you for the comments. We believe Section 6.4 contains empirical evidence that reduced parameter variance (quantitative results from Fig. 6, 10(b) and 10(c)) correlates with improved prediction stability and performance after model averaging.

---

### Decision · Action_Editor_QTn5 · 2026-02-27

**Recommendation:** Reject

**Audience:**

Yes

**Audience Explanation:**

Yes, weight averaging is widely used in practice, and the systematic comparison across architectures and datasets—particularly the finding that ViTs are substantially more fragile under weight averaging than CNNs—is practically useful and not well-documented elsewhere. Additionally, despite its shortcomings, the template-matching perspective could offer an accessible mental model and stimulate follow-up work.

**Claims And Evidence:**

No

**Claims Explanation:**

No, the claims are not sufficiently supported by the evidence. Thee of the four reviewers highlighted gaps between the framing and what the results actually demonstrate. The paper endeavors to explain why weight-averaging merging works, but the analysis mainly offers descriptive observations and visualizations rather than mechanistic explanations. The template-matching is validated only for linear classifiers, the links between variance and performance are not necessarily causal, and questions persist about various mathematical and implementation details. The reviewers also noted that the experimental scope is quite narrow, with only small-scale benchmarks and no comparisons to recent merging methods. While the rebuttal addressed some of these issues, the underlying gap between the claims and evidence remains.